# SLAPSHOT reveals rapid dynamics of extracellularly exposed proteome in response to calcium-activated plasma membrane phospholipid scrambling
Sami T. Tuomivaara [1,7,8], Chin Fen Teo [2,3,8], Yuh Nung Jan [2,3,4], Arun P. Wiita [1,5,6,9] ✉ & Lily Y. Jan [2,3,4,9] ✉

To facilitate our understanding of proteome dynamics during signaling events, robust workflows affording fast time resolution without confounding factors are essential. We present **S**urface-exposed protein **La**beling using **P**eroxida**S**e, **H**$_2$**O**$_2$, and **T**yramide-derivative (SLAPSHOT) to label extracellularly exposed proteins in a rapid, specific, and sensitive manner. Simple and flexible SLAPSHOT utilizes recombinant soluble APEX2 protein applied to cells, thus circumventing the engineering of tools and cells, biological perturbations, and labeling biases. We applied SLAPSHOT and quantitative proteomics to examine the TMEM16F-dependent plasma membrane remodeling in WT and TMEM16F KO cells. Time-course data ranging from 1 to 30 min of calcium stimulation revealed co-regulation of known protein families, including the integrin and ICAM families, and identified proteins known to reside in intracellular organelles as occupants of the freshly deposited extracellularly exposed membrane. Our data provide the first accounts of the immediate consequences of calcium signaling on the extracellularly exposed proteome.

The plasma membrane, whose structure and dynamics are integral to all aspects of cellular physiology, can accommodate rapid and drastic remodeling involving folding and unfolding[1] as well as fission and fusion[2–6]. Efforts to study the proteome-wide effects of such remodeling, especially in the context of signaling events, have been hampered by a general lack of suitable experimental approaches[7]. Established methods for labeling cellular proteins exposed to the extracellular milieu, such as sulfo-NHS-biotin[8] and cell surface capture (CSC)[9], have prohibitively long processing times for capturing these events. Experimental and confirmation biases toward known plasma membrane proteins have resulted in a scarcity of information regarding the full proteome exposed to the extracellular space. It is hence important to pursue technology development in this field.

Inspired by classical biochemistry and recent proximity labeling approaches, we report a method for rapid tagging of proteins exposed to the extracellular environment. Specifically, a soluble recombinant APEX2

enzyme[10] that generates short-lived phenoxyl radicals from phenolic substrates in the presence of H$_2$O$_2$, is employed. Applying APEX2 and its substrates on cellular preparations leads to rapid and covalent tagging of extracellularly exposed proteins. Useful substrates include biotin-tyramide (BT) and plasma membrane-impermeable biotin-xx-tyramide (BxxT), whose biotin handle can be used to enrich proteins targeted by the phenoxyl radicals. Our approach resembles peroxidase-mediated cell surface labeling (PECSL)[11] that employs a commercially available horseradish peroxidase (HRP) purified from plant tissue but is strategically different from other peroxidase-mediated methods where the enzyme is either surface-expressed[12] or anchored to the cell surface via additional binding motifs[13]. APEX2 neither depends on calcium binding nor contains disulfide bonds, rendering it a versatile tool for interrogating processes in the extracellular milieu.

We rigorously optimized the experimental conditions, aiming for a robust and reproducible method applicable to address diverse biological

---

[1]Department of Laboratory Medicine, University of California, San Francisco, CA, USA. [2]Howard Hughes Medical Institute, University of California, San Francisco, CA, USA. [3]Department of Physiology, University of California, San Francisco, CA, USA. [4]Department of Biochemistry and Biophysics, University of California, San Francisco, CA, USA. [5]Department of Bioengineering and Therapeutic Sciences, University of California, San Francisco, CA, USA. [6]Chan Zuckerberg Biohub San Francisco, San Francisco, CA, USA. [7]Present address: Department of Obstetrics, Gynecology & Reproductive Sciences and Sandler-Moore Mass Spectrometry Core Facility, University of California, San Francisco, CA, USA. [8]These authors contributed equally: Sami T. Tuomivaara, Chin Fen Teo. [9]These authors jointly supervised this work: Arun P. Wiita, Lily Y. Jan. ✉e-mail: arun.wiita@ucsf.edu; lily.jan@ucsf.edu

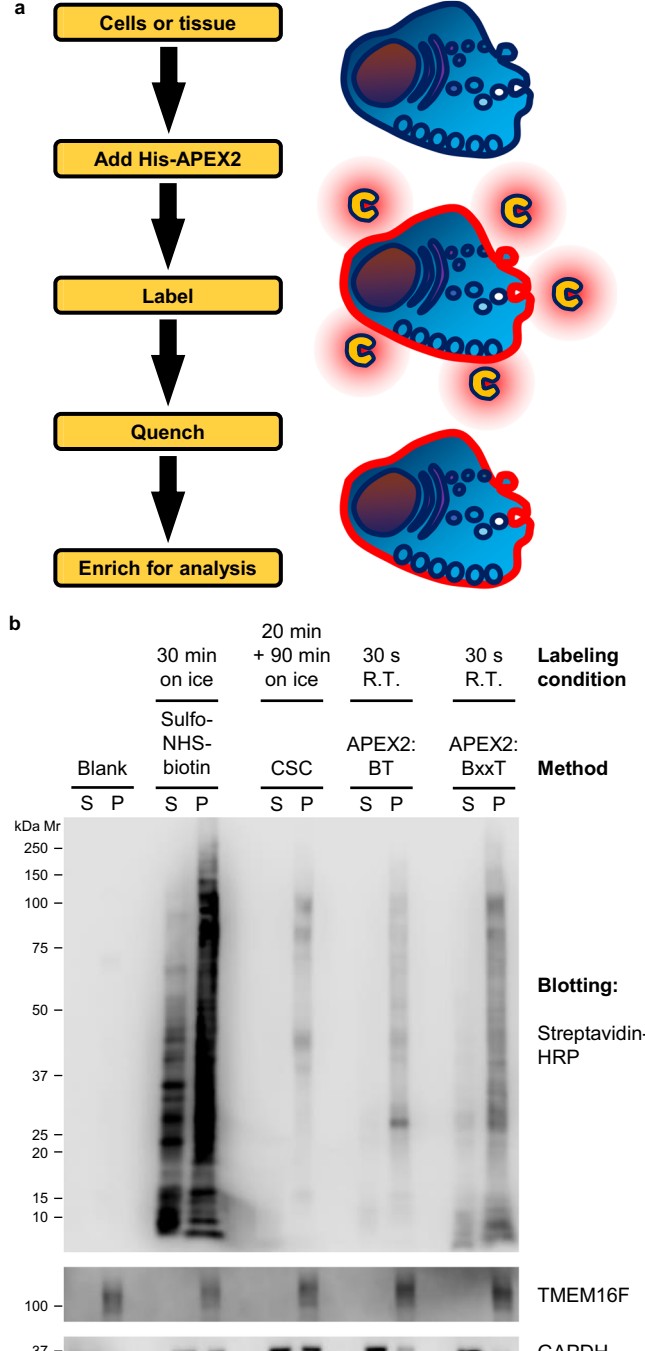

**Fig. 1 | SLAPSHOT effectively labels extracellularly exposed proteins of cultured cells. a** The SLAPSHOT workflow. Recombinant soluble APEX2 enzyme is applied to a biological sample, and the extracellularly exposed proteins are labeled in the presence of $H_2O_2$ and tyramide-derivative containing affinity handles such as biotin. Tagged proteins can be analyzed directly by streptavidin-blotting or enriched for MS or other analyses. **b** Streptavidin blot analysis of U2OS cells labeled with sulfo-NHS-biotin, Cell Surface Capture (CSC), and SLAPSHOT. After labeling, the cells were hypotonically lysed, crudely fractionated into soluble and insoluble fractions by centrifugation, and the samples were probed with streptavidin-HRP. The blotting indicates spatial selectivity of CSC and SLAPSHOT, as nearly all of the biotin signal is segregated in the membrane fraction together with plasma membrane-residing TMEM16F protein, rather than with the soluble GAPDH. SLAPSHOT was performed using either cell-permeable biotin-tyramide (BT) or cell-impermeable biotin-xx-tyramide (BxxT). Mr: relative molecular weight; kDa: kilodalton; S: soluble (cytoplasmic) fraction; P: insoluble pellet (crude membrane) fraction.

questions, and demonstrated that our method also proceeds rapidly under chilled conditions, limiting interference from confounding factors such as endo- and exocytosis. Having established the powerful, accurate, and fast delivery of this method, we named it SLAPSHOT for Surface-exposed protein Labeling using PeroxidaSe, $H_2O_2$ and Tyramide-derivative. Using SLAPSHOT, we investigated a curious phenomenon where the activation of the plasma membrane-residing calcium-activated phospholipid scramblase and non-selective ion channel TMEM16F[14,15] elicits rapid and massive dynamin 2-mediated cell surface expansion[16,17]. Interestingly, cells lacking TMEM16F display a drastic reduction in surface area upon this treatment[16]. Intriguing open questions include the proteomic composition of the recruited membrane and the full physiological ramifications of these phenomena. TMEM16F (TransMEMbrane protein family 16 member F, encoded by the *ANO6* gene) plays a critical role in extracellular vesicle (EV) release[18], plasma membrane wound repair[19], developmental and pathological syncytia formation[20,21], and host cell entry of pathogens including HIV[22], Ebola virus[23], and SARS-CoV[24]. TMEM16F is linked to Scott syndrome, a rare congenital bleeding disorder[14], where activated platelets from individuals with loss-of-function TMEM16F mutations fail to execute plasma membrane phosphatidylserine scrambling necessary for thrombin production and for the release of EVs that facilitate coagulation[25,26].

We applied SLAPSHOT and quantitative mass spectrometry (MS) to monitor the extracellularly exposed proteomes of wild-type (WT) and TMEM16F CRISPR-knockout (16FKO) Jurkat cells in response to calcium stimulation. SLAPSHOT-MS revealed differences in the extracellularly exposed proteomes of WT and 16FKO cells at resting state and upon calcium stimulation. We also detected synchronized regulation of protein complexes in the time-course analysis. Our MS data provides glimpses into calcium signaling and the dynamic interplay of plasma membrane and intracellular membranes. As an alternative and complementary approach to the available methods for labeling extracellularly exposed proteins, SLAPSHOT expands the toolbox toward fast signaling events. Our study serves as an example of using SLAPSHOT to empower investigations into the enigmatic and poorly understood dynamics of extracellularly exposed proteomes.

## Results
### SLAPSHOT: a rapid enzymatic workflow for tagging extracellularly exposed proteomes
To devise a method for rapid tagging of extracellularly exposed proteomes of living cells (Fig. 1a), we subcloned, expressed, and purified a soluble His-tagged APEX2 protein (Supplementary Fig. 1). This protein is stable in refrigerated conditions but sporadically loses activity when frozen (Supplementary Fig. 2a). We systematically evaluated labeling conditions on both adherent U2OS and suspended Jurkat cells by using streptavidin-blotting as a readout (Supplementary Fig. 3) and developed a colorimetric assay to gauge and standardize the peroxidase activity of APEX2 preparations (Supplementary Fig. 2b, c). We tested the effects of several parameters, including the enzymatic activity of APEX2, labeling duration, and the concentrations of BxxT and $H_2O_2$, on the labeling intensity (Supplementary Fig. 3a–c). Satisfactory incorporation of biotin to adherent and suspended cells can be achieved in 30 and 45 s, respectively, at room temperature in the presence of 0.0005 AU $s^{-1}$ $\mu L^{-1}$ APEX2 activity (see APEX2 colorimetric activity assay in "Methods"), 0.5 mM BxxT, and 0.5 mM $H_2O_2$. Throughout this work, we used a specified enzymatic activity of APEX2 rather than the molar amount for comparable labeling across biological samples and APEX2 preparations. The results from the optimizations of $H_2O_2$ concentration (Supplementary Figs. 2d, 3b) are consistent with the original characterizations of the APEX2 enzyme[27]. We also concluded that 10 mM Na-azide and 10 mM Na-ascorbate can fully quench the labeling reaction (Supplementary Fig. 2a, e).

We examined the spatial specificity of APEX2 labeling of U2OS cells, using either BT or BxxT as substrates, in parallel with sulfo-NHS-biotin[8] and CSC[9] methods. After hypotonically lysing and fractionating the cellular components into crude membrane and cytoplasmic fractions, we probed

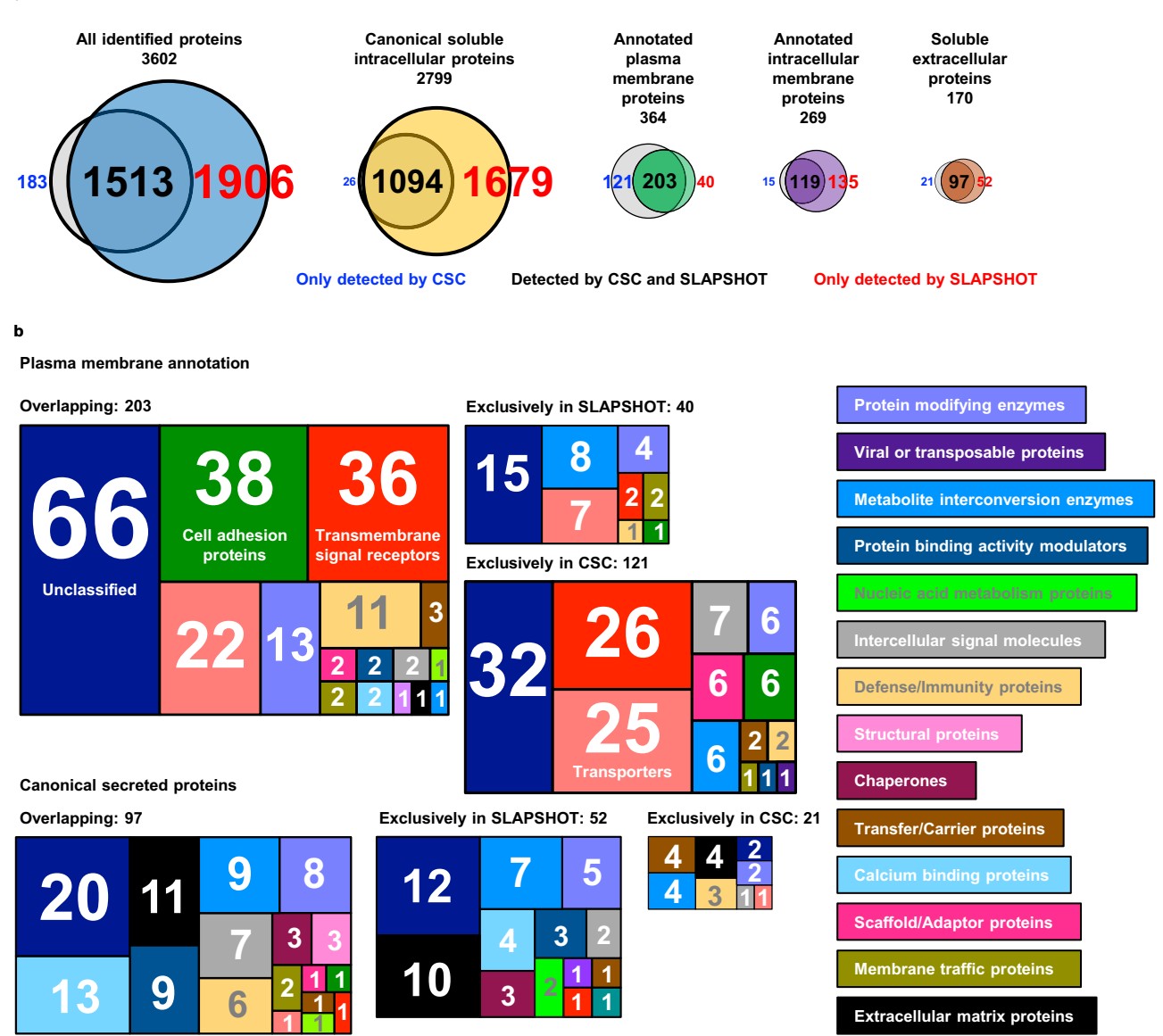

**Fig. 2 | The numbers and localization annotations of identified proteins in U2OS cells using SLAPSHOT and CSC. a** Venn diagrams indicate the number of identified proteins in total and with various localization annotations. In each diagram, the proteins identified by CSC are on the left, and proteins identified by SLAPSHOT are on the right. **b** PANTHER protein classes for the plasma membrane and secreted proteins. The total number of proteins, as well as proteins annotated in various compartments, are shown.

the fractions with streptavidin-blotting. Biotinylated materials from the APEX2 labeling segregate in the insoluble membrane-containing fraction with an established plasma membrane protein TMEM16F[14,15] (Fig. 1b, Supplementary Fig. 3d), indicating high spatial specificity of SLAPSHOT.

### SLAPSHOT does not compromise cellular integrity

Next, we performed a proof-of-principle MS-based proteomics analysis on U2OS cells comparing SLAPSHOT to CSC[9,28,29], the gold standard for cell surfaceome analyses. Using a label-free approach, we analyzed five replicate sets, each including cells treated with the two methods and a non-treated control. A total of 3602 proteins were identified, with 1513 (42.0% of total), 183 (5.1%) and 1906 (52.9%) detected by both methods, exclusively by CSC and exclusively by SLAPSHOT, respectively (Fig. 2a, Supplementary Data 1). When considering only proteins with plasma membrane (364 in total, based on consensus annotation from curated sources, see Statistical and bioinformatics analyses in Methods) and canonical secreted (170 in

total) annotations, the percentage of overlap increases slightly to 55.8% (203 IDs) and 57.1% (97 IDs), respectively (Fig. 2a).

Of proteins detected exclusively by CSC, 121 are plasma membrane-annotated and 21 are secreted, whereas of proteins detected exclusively by SLAPSHOT, 40 are plasma membrane-annotated and 52 are secreted. Of the 1814 proteins without plasma membrane or secreted annotation that were exclusively detected by SLAPSHOT, 1679 are annotated as soluble intracellular, and 135 are transmembrane proteins localized to various organelles. Notably, the soluble intracellular proteins include 1509 (89.9%) that were detected in the secretome and/or the EV proteome of U2OS cells[30] or have mouse homologs present in mitovesicles (EVs that contain mostly mitochondrially derived proteins)[31] (Supplementary Data 2). Of the 1120 soluble intracellular proteins from CSC analysis, 1094 are also detected by SLAPSHOT (Supplementary Data 2). Protein ANalysis THrough Evolutionary Relationships (PANTHER) protein classes of the SLAPSHOT-identified plasma membrane and secreted proteins include those involved in

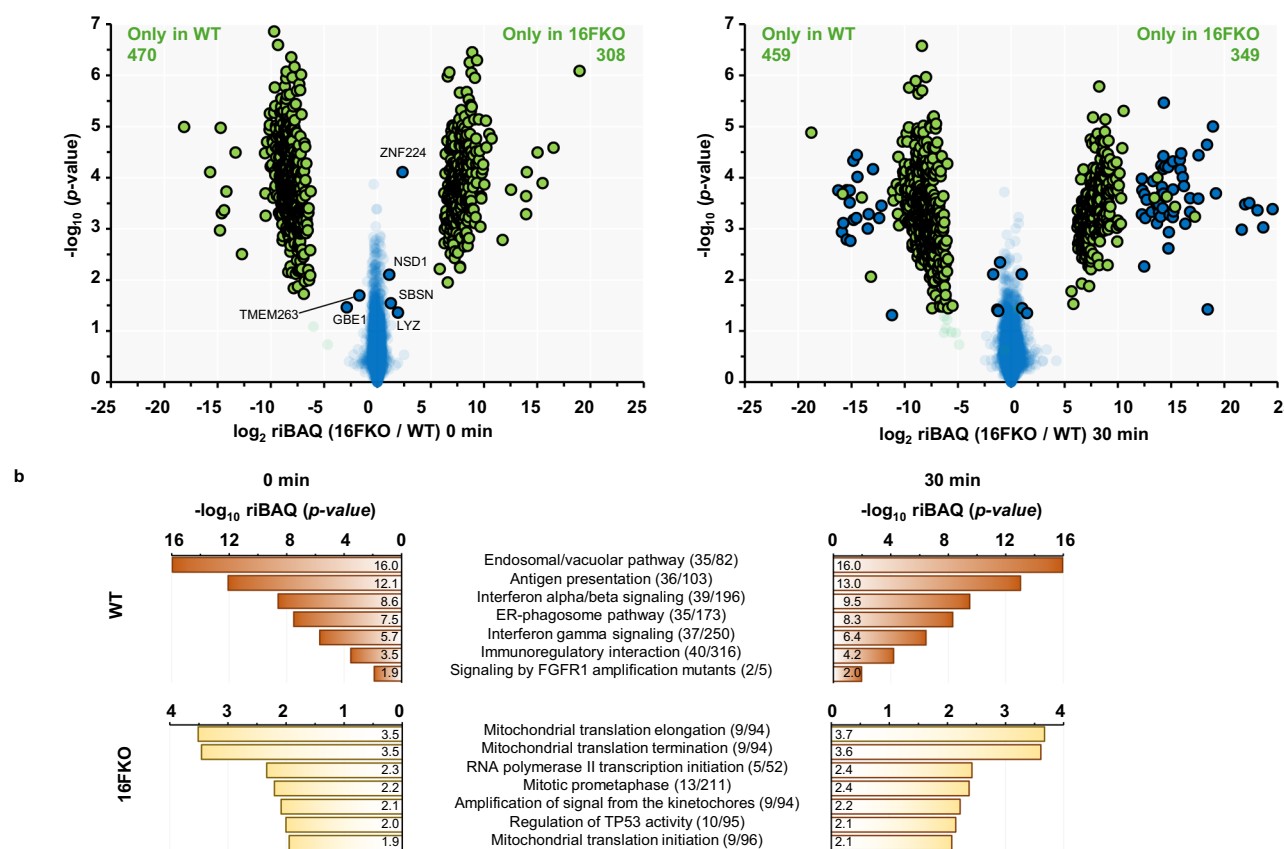

**Fig. 3 | SLAPSHOT-identified proteins from WT and 16FKO Jurkat cells at resting state and after 30 min of ionomycin stimulation.** Data from WT proteins are in left panels, and the 16FKO cells are in the right panels. **a** Volcano plots depict the proteomic differences between these two cell types for each cell type, $N = 3$. Blue data points indicate proteins that were identified in both samples. Data points in green were detected in only one of the samples and the other value was imputed. Circled data points have $\log_2$ transformed fold-difference of less than $-1$ or greater than 1 and are statistically significant ($p$-value < 0.05, Welch's test). **b** Reactome analysis of the proteins exclusively detected in the WT and 16FKO cells at the resting state and after 30 min of ionomycin stimulation. The pathways identified at 0 and 30 min are very similar in both cell types.

cell adhesion, transmembrane signal reception and transduction, as well as extracellular matrix scaffolding (Fig. 2b). SLAPSHOT and CSC samples cluster strongly based on the workflow (Supplementary Fig. 4), indicating distinct labeling targets. Collectively, our data suggest that the analytical depth of SLAPSHOT in capturing extracellularly exposed proteins is complementary to that of CSC.

Given the abundance of canonical intracellular proteins identified by SLAPSHOT, we investigated potential experimental artifacts by microscopy. Live-or-Dye staining profile of U2OS cells after room temperature SLAPSHOT was indistinguishable from that of untreated cells but drastically different from that of ethanol-perforated cells, indicating that SLAPSHOT preserves plasma membrane integrity (Supplementary Fig. 5a). We then examined the spatial selectivity of sulfo-NHS-biotin, CSC, and SLAPSHOT in Jurkat cells, by visualization of the biotin signal using fluorescently conjugated streptavidin. Biotinylation by all methods was restricted to the cell periphery (Supplementary Fig. 5b), albeit with differing intensities that likely resulted from labeling duration and target abundance differences. As revealed by our MS data, SLAPSHOT labels only Tyr and Trp residues (Supplementary Fig. 6) that have lower abundance and average solvent accessibility compared to Lys residues targeted by the NHS chemistry[32,33]. Furthermore, the CSC labeling intensity in microscopy is inflated by the numerous polymeric glycans decorating plasma membrane proteins[34] and by the abundant glycolipids[35]. The biochemical and microscopic data thus exclude the possibility of methodological artifacts and demonstrate the fidelity of SLAPSHOT in labeling living cells with improved time scales without compromising cellular integrity.

## SLAPSHOT permits temporally resolved analysis of signaling events using chilled cells to minimize vesicular traffic

Hilgemann and colleagues demonstrated that a brief 20 s calcium stimulation of wild-type (WT) Jurkat cells results in a rapid and massive cell surface area expansion, likely from dynamin 2 (*DNM2*)-dependent membrane unfolding, whereas TMEM16F deficient cells display an immediate reduction in the cell surface area by "massive endocytosis" (MEND)[16,17]. Little is known about the proteomic composition or the ultimate origin of the membrane, so we employed SLAPSHOT to address these questions. As a prerequisite, we established that SLAPSHOT proceeds in chilled conditions, where vesicular trafficking and endo- and exocytotic processes are largely arrested[36,37]. Signals similar to the level of room temperature labeling were obtained by labeling chilled adherent and suspended cells for 60 and 90 s, respectively (Supplementary Fig. 7).

We proceeded to treat WT and TMEM16F knockout (16FKO) cells, generated using CRISPR-Cas9 technology (Supplementary Fig. 8), with 1 μM ionomycin for 1, 5, 10, or 30 min at 37 °C to facilitate extracellular calcium entry into the cytoplasm (Supplementary Fig. 9), followed by SLAPSHOT. Triplicate MS data on the isobaric TMT10plex-labeled samples were collected for pre-stimulation (0 min, or resting state) control and each stimulation endpoint. A corresponding control without BxxT was included for each labeled sample. After background subtraction, we detected a total of 4148 proteins in WT and 16FKO cells (Fig. 3a, Supplementary Fig. 10, Supplementary Data 3). At resting states, 4138 proteins, of which 465 and 306 were exclusively detected in WT and 16FKO cells, respectively, were detected. Only six of the 3360 proteins detected in both cell types at the

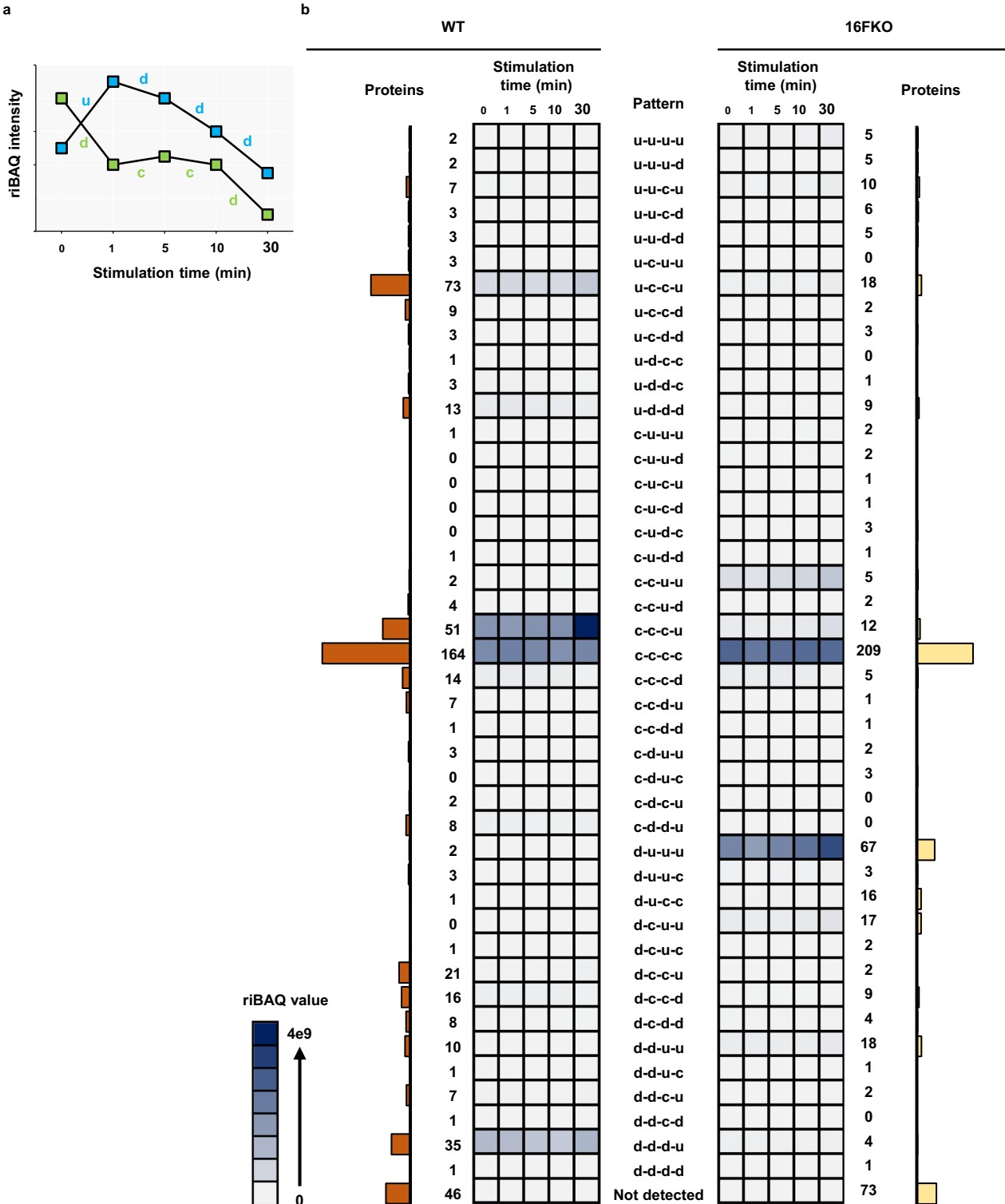

**Fig. 4 | Membrane proteins exhibit a wide variety of time-course patterns in their extracellularly exposed levels in response to ionomycin stimulation. a** Examples of time-course intensity patterns. **b** Heatmaps display the evolution of the total riBAQ for intensity patterns in WT (left heatmap) and 16FKO (right heatmap) cells when considering proteins with membrane annotation (N = 533). Each square in the grid represents the sum of riBAQ values from the proteins with that intensity pattern (rows) at that time point (columns). The total number of proteins belonging to a pattern from a given cell type is shown by numbers as well as the bar graphs. Notably, most of the proteins, as well as riBAQ intensity, are concentrated in just a few patterns.

resting state displayed significant differences in extracellular exposure level between the cell types (Fig. 3a). Reactome analysis of those proteins detected in WT cells at resting state and 30 min post-stimulation revealed enrichment in vesicular trafficking, antigen presentation, and immune-signaling

pathways (Fig. 3b). In contrast, extracellularly exposed proteins that were solely detected in the 16FKO cells are enriched in predominantly intracellular processes, such as those related to mitochondrial homeostasis. Only a small number of proteins displayed differential expression in response to

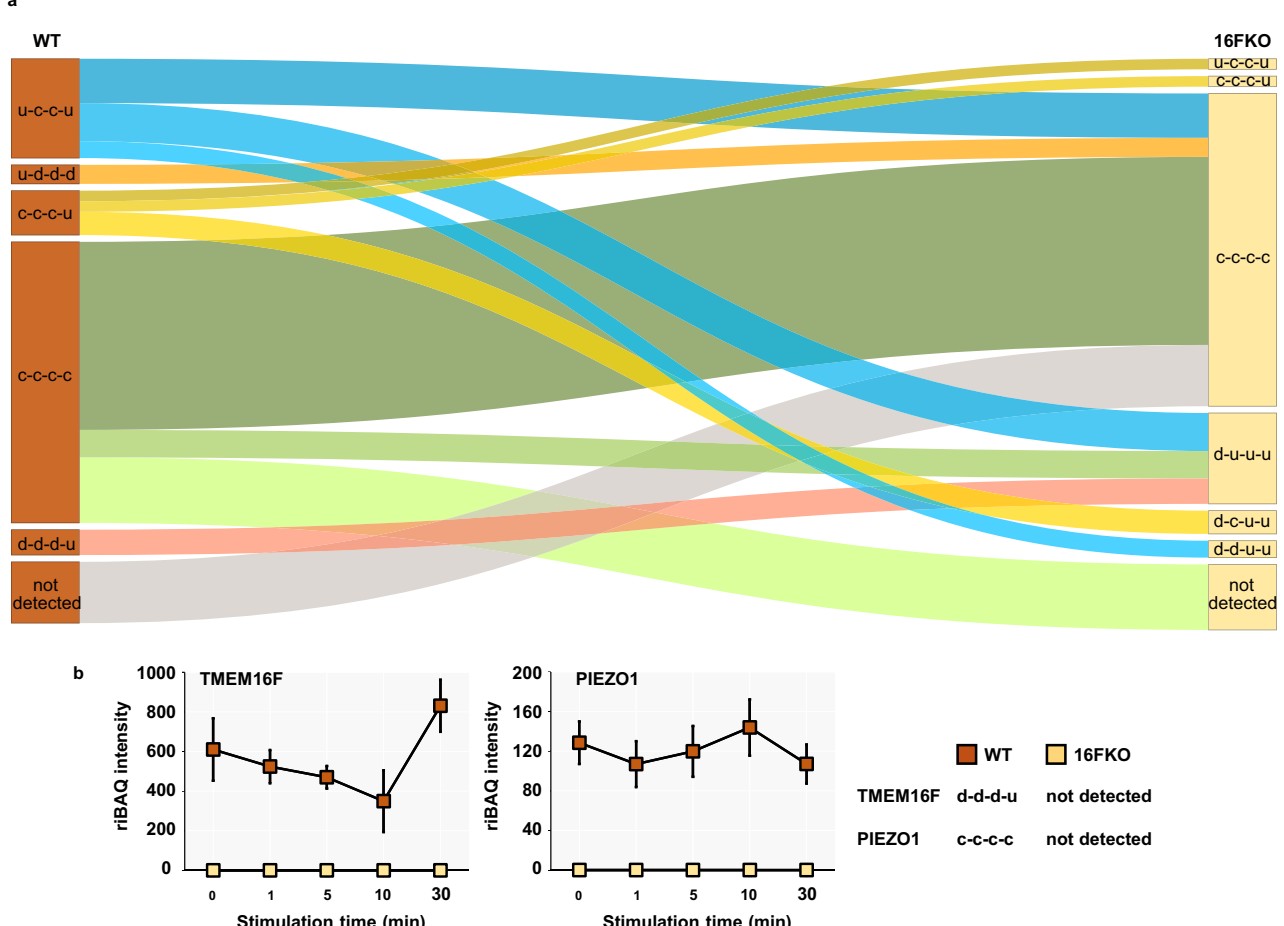

**Fig. 5 | Sankey flow diagram illustrates the shift of membrane proteins from one pattern in WT to another in 16FKO cells. a** Only the most prominent groups with number of proteins ≥ 5 (numbers indicated), and *p*-value < 0.05 for either over- or underrepresentation compared to random assortment (Fisher's exact test) are shown. WT u-c-c-u to 16FKO c-c-c-c and WT c-c-c-c to 16FKO d-u-u-u contain less proteins than expected (*p*-value < 0.05, Fisher's exact test), whereas all other transitions contain more proteins than expected. **b** Time-course intensity patterns of TMEM16F and PIEZO1 in response to calcium stimulation.

calcium stimulation at 10 min or earlier time points, compared to the pre-stimulated control, in either cell type (Supplementary Fig. 10 and Supplementary Data 4). The reproducibility of the multiplexed isobaric analyses is manifested by hierarchical clustering, whereby the samples segregate strongly by the triplicate analyses and the cellular genotype (Supplementary Fig. 11).

## TMEM16F-dependent dynamics of the extracellularly exposed proteome

To systematically investigate the trends and diversity in the extracellularly exposed status of the detected proteins, we fitted their riBAQ intensities against a collection of models (Supplementary Data 5) and classified each protein according to the best-fitting model. We assigned a single letter to represent the direction of the intensity change between two consecutive time points, such that "c", "u", and "d" respectively represent "constant", "up", and "down" (Fig. 4a). For instance, a protein whose intensity decreases over the first interval (from 0 to 1 min) and increases over the other three intervals (1 to 5, 5 to 10, and 10 to 30 min) is classified as "d-u-u-u". Of the 61 theoretical patterns, 56 are represented by at least one detected protein (Supplementary Fig. 12, Supplementary Data 5). Surprisingly, only 1167 (28.2%) of the 4148 detected proteins have the same pattern in both cell types (Supplementary Data 6). As the riBAQ values indicate the protein relative molar amount both within and between samples, we tabulated the total riBAQ values across time points and patterns in a heatmap (Supplementary Fig. 12). The majority of proteins are assigned into just a few patterns, with most of them occupying the c-c-c-c pattern in both cell types.

We next performed analyses focusing solely on proteins with trans-membrane and/or GPI-anchor domains ("membrane proteins", based on consensus annotation from curated sources). Overall, 488 and 460 membrane proteins were respectively identified in WT and 16FKO cells, populating 43 distinct patterns. Six patterns are prominent with at least 5% of all detected membrane protein IDs or 5% of the total riBAQ intensity (Fig. 4b). We identified patterns where their fractional riBAQ intensity (of total riBAQ intensity in the experiment) is either significantly smaller or significantly larger compared to the fraction of identified membrane proteins that shows a given pattern (Fig. 5a). A single pattern in WT cells, u-c-c-u, belongs to the former category: its 73 occupants (15% of identified membrane proteins) contribute only 8.3% of the total riBAQ intensity. Four patterns, c-c-c-u and d-d-d-u in WT cells, and d-u-u-u and c-c-u-u in 16FKO cells, exhibit the inverse trend: 51 c-c-c-u WT membrane proteins (10.5%) with 33.9% riBAQ, 35 d-d-d-u WT membrane proteins (7.2%) with 14.9% riBAQ, 67 d-u-u-u 16FKO membrane proteins (14.6%) with 34.5% riBAQ, and 5 c-c-u-u 16FKO membrane proteins (1.1%) with 7.8% riBAQ intensity.

We did not detect TMEM16F in the 16FKO cells as expected, but did so in all WT cell time points, with the d-d-d-u pattern (Fig. 5b). The ratios of APEX2 intensity between the labeled and corresponding control samples do not differ significantly among the pre-stimulated and any of the calcium-stimulated samples, lending credence to the notion that the membrane-impermeable reagents, short labeling time, and chilled conditions limit reagent internalization by calcium-dependent endocytosis or any other process (Supplementary Fig. 13). We attribute the constant and mere 2.5-

fold higher riBAQ intensities in the labeled samples compared to the non-labeled controls to a visible colloid possibly containing APEX2 cross-linked with polymerized BxxT. We also detected transferrin receptor (*TFRC*), CD28 and various human leukocyte antigen (HLA) isoforms previously detected by Bricogne et al.[16] in a flow cytometry experiment using programmed cell death protein 1 (PD-1, *PDCD1*) overexpressing WT and 16FKO Jurkat cells after 15 min of ionomycin stimulation (Supplementary Fig. 14).

Further scrutiny of the intensity patterns yielded the following insights

(a) Proteins with constant intensity (c-c-c-c pattern) are the largest group of membrane proteins in both cell types, with 164 (34%) and 209 (45%) proteins in WT and 16FKO cells, respectively, accounting for 31% and 39% of overall riBAQ intensities. Only 89 of the membrane proteins with this pattern are detected in both cell types. Mechanosensitive ion channel piezo-type mechanosensitive ion channel component 1 (*PIEZO1*) is one of the 31 membrane proteins with c-c-c-c pattern in WT cells that are not detected in 16FKO cells (Fig. 5b). It is noteworthy that the extracellularly exposed PIEZO1 protein levels in WT cells are very low as per the riBAQ values but can be consistently detected by SLAPSHOT-MS. Due to the lack of anti-PIEZO1 antibodies that work unambiguously in Jurkat cell lysates, we examined the expression level of PIEZO1 transcript and found no difference between WT and 16FKO cells (Supplementary Fig. 15).

(b) Membrane proteins with d-d-d-u pattern in WT cells include those in the integrin and the intercellular adhesion molecule (ICAM) families (Fig. 6a), some of which are known to physically interact with each other[38]. A tight regulatory relationship is further indicated by their co-regulation in 16FKO cells with the d-u-u-u pattern (see below). Given that EVs, along with their constituent proteins, separate from the plasma membrane within the first 10 min[16] of stimulation in a TMEM16F-dependent manner[18,19,39–41], the d-d-d-u pattern could reflect this process following with recovery to pre-stimulation levels. This interpretation is bolstered by the finding that TMEM16F itself followed this pattern in WT cells (Fig. 5b).

(c) Massive endocytosis (MEND), whereby parts of the plasma membrane are acutely internalized upon calcium stimulation, is the primary membrane remodeling process in 16FKO Jurkat cells[16]. We speculate that 16FKO membrane proteins with d-u-u-u pattern (Fig. 6b) reflect MEND for several reasons. First, the immediate reduction in the protein intensity, followed by their restoration, matches the time course and physical description of MEND. Second, d-u-u-u is the second largest pattern for membrane proteins in 16FKO cells based on both the number of IDs and total riBAQ intensity, suggestive of MEND, where a large portion of the plasma membrane is internalized. Just two proteins with this pattern were detected in the WT cells and neither of them displays the d-u-u-u pattern in 16FKO cells, in line with the uniqueness of MEND to the 16FKO cells (Supplementary Data 6). Plasma membrane-associated GO cellular component annotations of these proteins are consistent with this hypothesis (Supplementary Fig. 16).

(d) Proteins with u-c-c-u pattern in the WT cells account for the majority of the freshly deposited proteins during the rapid cell surface area expansion[16] for the following reasons. First, in WT cells, u-c-c-u is the only pattern with an immediate increase in intensity that contains a significant number of constituent membrane proteins (Fig. 4b). Second, PANTHER GO cellular component annotations of these proteins indicate a residency in ER and other intracellular membranes (Supplementary Fig. 16). Third, their collective riBAQ value is significantly lower than the average membrane protein (8.3% of total riBAQ versus 15.0% of protein IDs), suggesting a non-constitutive presence on the plasma membrane. These data are consistent with the observations whereby the newly deposited membrane is nearly devoid of electrophysiological activity and likely contains little protein compared to the resting plasma membrane[16]. Fourth, the levels of most of the proteins that display the u-c-c-u pattern in the WT cells are either unchanged (c-c-c-c) or show immediate reduction (d-u-u-u, or d-d-d-u) in their intensities in 16FKO cells (Figs. 4a, 6c).

(e) The c-c-c-u pattern in WT cells is the third most populated by the number of proteins, and the most populated by their riBAQ intensity

(Fig. 4b). This group contains roughly equal number of ER and plasma membrane proteins, and curiously, mitochondrial proteins as well (Fig. 6d). The increase of ER and plasma membrane proteins suggest an ongoing membrane restoration after continuously elevated calcium flux. The majority of the mitochondrial proteins detected in our assay are also found in mitovesicles[31]. Our SLAPSHOT data suggests that mitovesicle release does not follow the trajectory of canonical EVs whose release depends on TMEM16F activation.

(f) The c-c-u-u pattern in 16FKO cells only contains five proteins but accounts for a higher riBAQ value than many other patterns with more protein constituents (Fig. 6e). The major contributor to the riBAQ value is voltage-dependent anion-selective channel protein 3 (*VDAC3*), also a constituent of mitovesicles[31]. Prompted by this finding, we stratified the overall riBAQ values of all time points based on the cellular compartment and found high levels of proteins from mitovesicles throughout the stimulation (Supplementary Fig. 17). In WT cells, their riBAQ intensities are relatively constant following ionomycin stimulation until an increase at the 30 min time point, whereas in 16FKO cells, total riBAQ level of mitovesicle proteins increased slightly in earlier time points.

We also used western blotting to probe for selected proteins in the isolated EVs as well as in the corresponding cell pellets from WT and 16FKO Jurkat cells, after ionomycin stimulation (Supplementary Fig. 18). We confirmed that TMEM16F is released from the WT cells throughout the stimulation, and conversely, decreasing amounts were left in the cells, indicating that a substantial portion of TMEM16F is localized to the membrane domains that are shed into extracellular vesicles. We also detected a steady release of α-tubulin (*TUB1A1*) and glyceraldehyde-3-phosphate dehydrogenase (*GAPDH*)—two known extracellular vesicle proteins—as well as acyl-CoA synthetase long-chain family member 1 (*ACSL1*), a known mitovesicle protein, consistent with the role of TMEM16F in the EV release. Substantially higher amounts of all these proteins were found in the WT cells compared to the 16FKO cells (Supplementary Fig. 18), in agreement with our MS analyses. As a negative control for cellular contamination in the EVs, we also probed for ER-residing diacylglycerol O-acyltransferase 2 (*DGAT2*), which has never been reported in EVs (as per Vesiclepedia). Importantly, we did not detect DGAT2 by SLAPSHOT-MS, further bolstering the case for its spatial selectivity toward extracellularly exposed proteins, including those presenting in EVs.

## Discussion
We present SLAPSHOT, a protein tagging approach that utilizes soluble recombinant APEX2 enzyme to rapidly label extracellularly exposed proteins in living cells. After extensive optimizations, we conclude that SLAPSHOT compares favorably with the CSC[28] and sulfo-NHS-biotin[8] approaches. First, the rapid APEX2 kinetics allows the capture of ephemeral signaling processes, a feat not practicable with known chemistry-based methods, including CSC, whose total workflow duration approaches 2 h. Second, Tyr and Trp residue labeling is less likely to interfere with the downstream workflow[42,43], for instance, as in the case of CSC, where the ubiquitous glycolipids on the plasma membrane compete for the enrichment capacity of the solid support. Third, direct protein tagging accurately reflects the protein abundance, unlike methods relying on unpredictable proxy measurements whose fluctuations between samples may overwhelm any underlying proteome changes. Aberrant post-translational modifications, including glycosylation and carbonylation, often accompany cellular pathologies[44,45] and may affect CSC labeling, while non-glycosylated proteins exposed to the extracellular milieu may escape detection via CSC.

SLAPSHOT also presents improvements compared to other peroxidase-based surfaceome labeling methods[11–13]. Unlike the recently reported approaches where APEX2 is physically tethered to the plasma membrane through an engineered binder[13] or where HRP is fused to a plasma membrane protein[12], the soluble APEX2 employed in SLAPSHOT is free of any spatial constraints, and thus free from protein or membrane microdomain-specific biases. SLAPSHOT does not require tedious

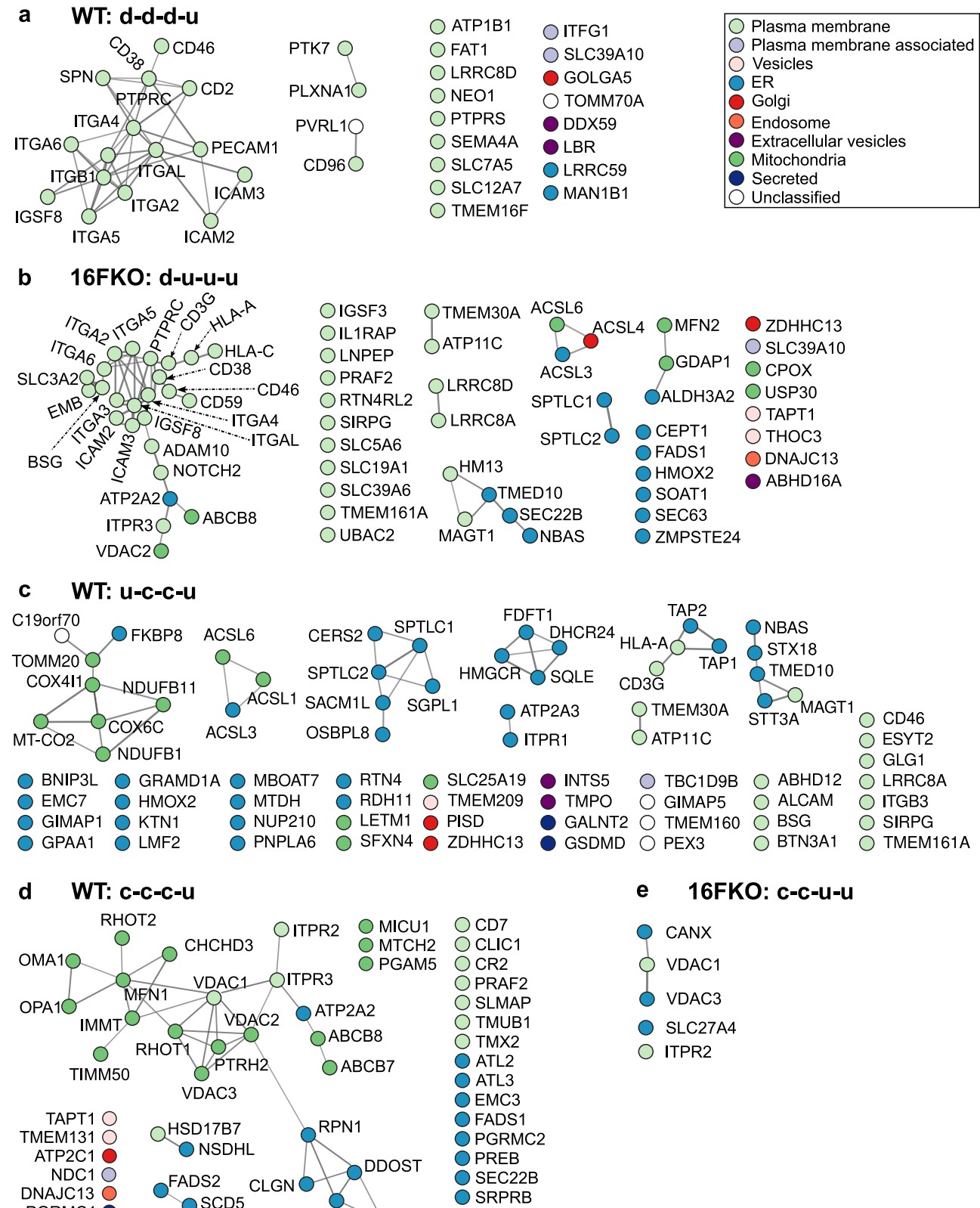

**Fig. 6 | STRING analysis of membrane proteins sharing a time-course intensity pattern. a** WT d-d-d-u. **b** 16FKO d-u-u-u. **c** WT u-c-c-u. **d** WT c-c-c-u. **e** 16FKO c-c-u-u. While plasma membrane proteins dominate most of these patterns, many ER and/or mitovesicle proteins also contribute to WT u-c-c-u and WT c-c-c-u patterns.

adaptations to various biological models and thus can be used for rare and primary cells as well as ex vivo tissue samples that are refractive to genetic manipulations. Unlike HRP utilized in PECSL[11], APEX2 is neither metal-dependent nor redox-sensitive. Importantly, we established that the labeling

efficiency of SLAPSHOT is not prohibitively hampered by ice-cold conditions (Supplementary Fig. 7), allowing investigations into proteome dynamics without confounding effects from cellular processes such as endocytosis. While we have not performed cell number titrations, we

collected reproducible MS data from a single 10 cm dish of U2OS cells and from 5 million suspended Jurkat cells, demonstrating the sensitivity and potential utility of SLAPSHOT in labeling scarce cell populations.

The role of calcium signaling in cellular physiology is well appreciated[46], but little is known about its effects on the plasma membrane proteome dynamics. Toward a greater understanding of this essential signaling mechanism, we applied SLAPSHOT coupled to a quantitative MS-based proteomics workflow to interrogate TMEM16F-dependent plasma membrane remodeling upon calcium stimulation in WT and 16FKO Jurkat cells. SLAPSHOT-MS allowed reconstruction of the dynamics of 4148 extracellularly exposed proteins, of which 533 are membrane proteins (including most of the proteins assessed by Bricogne et al.[16], Supplementary Fig. 14), after calcium stimulations as short as 1 min. At the resting state, many detected proteins in WT, but not on 16FKO cells, are involved in antigen presentation and T cell receptor signaling (Fig. 3). Our data agree with previous reports implicating TMEM16F in T cell receptor-mediated signaling[47,48]. Moreover, for proteins detected in both WT and 16FKO cells, more than 86% of these extracellularly exposed proteins have different time-course intensity profiles in cells with or without TMEM16F (Supplementary Data 6). These data highlight the fluidity of the global protein network and the complex consequences of removing the product(s) of a single gene from a biological system.

The lack of flow cytometry- and microscopy-compatible TMEM16F antibodies has hindered direct examination of its surface expression. SLAPSHOT allowed the observation of a gradual decrease in its surface levels before recovery to pre-stimulation level (Fig. 5b). Immunoblotting data (Supplementary Fig. 18) indicates that the decrease of extracellularly exposed TMEM16F is not due to internalization, but rather from its shedding with EVs, a pattern that reflects its intimate involvement in EV release[18,19,39–41]. Besides ionomycin stimulation, activation of TMEM16F and subsequent cellular surface area expansion can be triggered by mechano-sensing ion channel PIEZO1 in hypotonic medium conditions[16]. We did not detect PIEZO1 in 16FKO cells by MS (Fig. 5b), even though we did detect its transcript at comparable levels to that in WT cells (Supplementary Fig. 15), indicating that TMEM16F impinges on PIEZO1 surface expression. Further studies are needed to decipher the regulatory feedback mechanisms involving TMEM16F and PIEZO1.

Our study opened an avenue for deciphering protein contributions to the calcium signaling-triggered plasma membrane expansion in WT cells and the massive endocytosis (MEND) in 16FKO cells. In WT cells, we identified a subset of proteins that are known to also reside in the ER, whose u-d-d-d pattern matches that of the dynamin 2-dependent membrane unfolding (Supplementary Fig. 12). ER and plasma membrane contact sites are well known to participate in cellular calcium signaling and homeostasis[49,50]. Here we provide new evidence that proteins with ER localization can be exposed to the extracellular milieu upon calcium stimulation.

In 16FKO cells, we identified many plasma membrane proteins with d-u-u-u pattern suggesting their removal from the cell surface by MEND immediately following calcium stimulation (Fig. 6b, Supplementary Fig. 12). Furthermore, we found that a cluster of cell adhesion proteins from the integrin and ICAM families exhibit d-d-d-u pattern in WT cells following calcium stimulation but display d-u-u-u pattern in 16FKO cells (Figs. 5a, 6a, b). The synchronous change in the extracellular exposed pattern of these known protein complexes[38] further inspires confidence in and supports the accuracy of the SLAPSHOT method.

SLAPSHOT-MS yielded more canonical intracellular proteins compared to CSC and a unique set of proteins with plasma membrane annotation but not detected by CSC (Fig. 2, Supplementary Data 1, 2). The sizable fraction of intracellularly annotated proteins labeled in U2OS cells by both SLAPSHOT and CSC, with orthogonal chemistries and targets, also suggests that plasma membrane proteins are merely a subset of the total extracellularly exposed proteome (Supplementary Data 2). The CSC chemistry was designed to target glycoproteins on the plasma membrane but can chemically target mainly intracellular *O*-GlcNAcylation[51] and

carbonylation[52], which are intrinsically low-stoichiometry post-translational modifications. That said, research questions focusing strictly on plasma membrane proteins and where time resolution is not a concern could benefit from CSC, which is less affected by intracellular protein labeling compared to SLAPSHOT. It remains an open question as to how the proteome coverage may be affected by the lower abundance and accessibility of Tyr and Trp residues, compared to Lys residues and glycan diols with different distributions in proteins.

In all our SLAPSHOT-MS experiments, we detected a large number of proteins residing in mitovesicles, a newly discovered EV whose release mechanism remains elusive but which could be affected in pathological conditions[31,53]. While the release machinery for mitovesicles remains unclear, their distribution in the SLAPSHOT-MS dataset suggests it only partially relies on TMEM16F activation. We surmise that SLAPSHOT, without any ultracentrifugation steps, could be a useful approach to enrich mitovesicles and other EVs.

SLAPSHOT-MS yields quantitative and reproducible data with high time resolution and does not require genetic engineering of the target cells or individualized detection tools. In light of the technical maturity of quantitative MS-based proteomics, one may consider dispatching MS data validation by orthogonal detection methods[54,55]. As immunoaffinity-based fluorescence-assisted cell sorting and microscopy approaches that are often used for validating, or in lieu of, MS, do not offer the same combination of time resolution and "high-plexity" as SLAPSHOT-MS, we echo these notions. Of the available MS strategies, we opted for data-dependent acquisition (DDA) and MS[1]-based quantitation. Furthermore, for the calcium stimulation experiments, we utilized isobaric TMT labeling due to the multiplexing and precision offered by the stable isotope labeling. As DDA approaches suffer from stochastic peptide coverage between injections, data-independent acquisition (DIA) approaches are an attractive alternative, given the reproducible peptide coverage they offer[56].

In summary, we present an enzyme-based tagging method for extracellularly exposed proteins with fast kinetics that can monitor rapid proteome dynamics in a systematic and unbiased manner. Collectively, our data indicate that SLAPSHOT complements the existing labeling approaches. Given its simplicity and flexibility, SLAPSHOT can be readily adapted for studies that examine the dynamics of extracellularly exposed proteins in various biological contexts. Intriguing prospects for SLAPSHOT include the determination of cell surface processes during various signaling events[1,3,47] and membrane wound healing[19].

## Methods
All reagents were purchased from Sigma-Aldrich unless otherwise noted.

### Cloning, heterologous expression, and purification of hexahistidine-APEX2
We amplified the open reading frame encoding the engineered ascorbate peroxidase APEX2 from APEX2-NLS plasmid (Addgene plasmid #49386, a kind gift from Dr. Alice Ting) with Q5 polymerase (New England Biolabs) and subcloned the amplicon onto PCR linearized pRESTc vector (Invitrogen) with In-Fusion assembly reagent (Takara Bio). The resulting plasmid, pRESTc-His$_6$-APEX2, was confirmed with Sanger sequencing and transformed into T7 Express *lysY* competent *E. coli* for recombinant protein expression. A single colony picked from a carbenicillin (carb)-containing LB plate was grown in LB/carb broth overnight (37 °C, 250 rpm) and subcultured into Terrific Broth/carb (37 °C, 250 rpm) to an OD$_{600}$ of 0.8 to 1.2. His-APEX2 protein expression was induced with IPTG at 0.5 mM, and the bacteria were harvested 2 to 3 h post-induction. We resuspended the bacterial pellet in 150 mM NaCl, EDTA-free HALT protease inhibitor cocktail (Thermo Scientific), 1 mM phenylmethylsulfonyl fluoride (PMSF), 1 mg/mL lysozyme, 50 mM Tris-HCl (pH 7.5) lysis buffer and incubated it at room temperature for 30 min with occasional vortexing. We then added MgCl$_2$ to a final concentration of 10 mM and DNAse I (>500 Kunitz U/mg, GoldBio) to a final concentration of 1 mg/mL and incubated the lysate for an additional 15 min at room temperature. From here on, the APEX2

preparations were handled either at 4 °C or on ice. After clarifying the lysate by centrifugation (12,000×$g$, 15 min), we mixed the supernatant with pre-washed high-density cobalt-agarose resin (GoldBio) for batch binding. After 1 h incubation with gentle mixing, the resin was poured into an empty fritted column. We washed the resin extensively with 150 mM NaCl, 10 mM imidazole, 50 mM Tris-HCl (pH 7.5) buffer, eluted resin-bound APEX2 with 150 mM NaCl, 500 mM imidazole, 50 mM Tris-HCl (pH 7.5) buffer and concentrated the eluate with a pre-washed 10 kDa MWCO Amicon Ultra spin filter (Millipore). The concentrated eluate was buffer-exchanged into 20 mM K-phosphate (pH 7.0) using a pre-washed PD10 size-exclusion column (Bio-Rad) using gravity flow. We modified a reported procedure[57] to prepare a saturated hemin solution. Briefly, hemin-Cl was suspended in 10 mM NaOH to 25 mg/mL concentration by vigorous vortexing, diluted 5-fold in 20 mM K-phosphate (pH 7.0), and cleared by centrifugation (3000×$g$, 10 min). Hemin solution was mixed with the APEX2 sample for 3 h on ice (from here on, the sample was kept protected from light to prevent photodamage). We next applied the hemin reconstituted APEX2 to pre-washed DEAE-Sepharose resin slurry (Cytiva) and incubated for 30 min to capture the excess hemin. The slurry was washed extensively with 20 mM K-phosphate (pH 7.0) buffer, followed by elution of APEX2 with 100 mM K-phosphate (pH 7.0) buffer. Finally, we concentrated the APEX2 preparation with a pre-washed 10 kDa MWCO Amicon Ultra spin filter, sterile filtered it through a 0.2 μm filter, and stored it at 4 °C protected from light. Final APEX2 concentration was measured with a BCA assay (Thermo Scientific) using BSA (Fisher Scientific) as a standard, and its densitometric purity was determined in Fiji[58]. In contrast to previous reports[13,27], we did not freeze the purified protein, as this destroyed the enzymatic activity in our hands, and utilized the refrigerated APEX2 within four weeks of purification.

## APEX2 colorimetric activity assay
We developed a colorimetric assay for the APEX2 peroxidase activity, based on the oxidation of catechol to *ortho*-quinone in the presence of $H_2O_2$. According to a wavelength scan, the reaction products absorb maximally at 480 nm in the experimental conditions outlined below. We mixed 96 μL of 1 M glycine (a sink for the very reactive *ortho*-quinone[59] that would otherwise inhibit APEX2 enzyme), 100 mM Tris-HCl (pH 7.5) buffer with 2 μL of appropriately diluted APEX2 and 2 μL of freshly made aq. 100 mM catechol solution onto a 96-well microplate. To this solution, we mixed 100 μL of 1 mM $H_2O_2$ (obligate electron acceptor), 1 M glycine, 100 mM Tris-HCl (pH 7.5) buffer, and immediately measured absorbance at 480 nm using a Synergy 2 or Synergy H4 microplate reader (BioTek), at 2 s intervals. The activity of the APEX2 solution equals the slope of the absorbance-vs-time curve (AU s$^{-1}$ (2 μL)$^{-1}$) for the early linear portion where Pearson correlation ≥0.998 (usually no more than 30 s when concentrated APEX2 solutions were used). The activity of the original APEX2 preparation (AU s$^{-1}$ μL$^{-1}$) was calculated by dividing the slope by the volume used in the activity assay (typically 2 μL), and by multiplying by any prior APEX2 dilution factor from the stock. We ascertained the linearity of the activity assay using a serial dilution of APEX2 solution. We also tested the effects of $H_2O_2$ concentration and quenchers (Na-azide, Na-ascorbate) doped in the $H_2O_2$ solution, as well as storage conditions, on the APEX2 activity. Blank experiments were conducted by omitting ether catechol or $H_2O_2$. The enzymatic activity of refrigerated APEX2 preparations was measured each day before the labeling experiments.

## APEX2 turbidimetric activity assay
We developed a turbidimetric assay for the APEX2 peroxidase activity, based on the formation of a visible colloid in all APEX2 reactions with BT or BxxT substrate. According to the wavelength scan, the reaction products scatter maximally at 400 nm in the experimental conditions outlined below. We mixed 97 μL of 100 mM Tris-HCl (pH 7.5) buffer with 2 μL of APEX2 and 1 μL of 100 mM BT (APExBIO) in DMSO onto a 96-well microplate. To this solution, we then mixed 100 μL of 1 mM $H_2O_2$, 100 mM Tris-HCl (pH 7.5) buffer and immediately measured the optical density at 400 nm using a Synergy 2 or Synergy H4 microplate reader (BioTek), typically at 2 s intervals for 1 min. We also tested the effects of quenchers (Na-azide, Na-ascorbate, Trolox) by doping them in the $H_2O_2$ solution. Blank experiments were conducted by omitting either BT or $H_2O_2$.

## Cell culture
We cultured all cells in humidified incubators at 37 °C with 5% $CO_2$. Adherent U2OS cells (from Cell and Genome Engineering Core, UCSF) were grown in DMEM media (Corning) supplemented with 10% FBS (Axenia BioLogix). The cells were seeded respectively with 2.5e5 cells on 3.5 cm (for optimizing conditions), or 1e6 cells for 10 cm dishes (for proteomics) 48 h prior to experimentation and grown to ~90% confluency. The Jurkat E6.1 T cells (a gift from Dr. Arthur Weiss, UCSF) were maintained in suspension at 0.2e6 to 1e6 cells/mL cell density either in regular RPMI1640 media supplemented with 10% FBS (Atlanta Biologicals) (for optimizing conditions), in advanced RPMI1640 medium (Life Technologies) supplemented with 2% in-house heat-treated (56 °C for 30 min) FBS (Atlanta Biologicals) and GlutaMAX (Gibco) (for proteomics), or in advanced RPMI1640 medium supplemented with 2% extracellular vesicle-depleted FBS (Gibco) and GlutaMAX (for extracellular vesicle isolation). AMO1 cells (a gift from Dr. Christoph Driesen, Kantonsspital St. Gallen, Switzerland) in suspension were cultured at 1e6 to 2e6 cells/mL cell density in RPMI 1640 medium supplemented with 1% penicillin/streptomycin (ThermoFisher Scientific) and 20% FBS (Atlanta Biologicals). We confirmed mycoplasma negativity for all cultures prior to experiments (DigitalTest v2.0, bio-tool.com). Replicate experiments were performed on cells from different passages.

## Generation of TMEM16F knockout cell line
We created a clonal TMEM16F (encoded by the *ANO6* gene) CRISPR-knockout (KO) Jurkat cell line using the Alt-R CRISPR-Cas9 system (Integrated DNA Technologies, IDT) by following the manufacturer's protocol. Briefly, we annealed two crRNAs both targeting human ANO6 exon 6 (crRNA1 sequence: UGU AAA AGU ACA CGC and crRNA2 sequence: CUG AAA AAC CGG UCC, 3 μL of 200 μM each, IDT) independently with tracrRNA (3 μL of 200 μM, IDT) and mixed each of the RNA complexes with purified Cas9-NLS (12.2 μL of 41 μM, QB3 Berkeley, University of California) to obtain two tubes of RNPs each with a final volume of 20 μL. We prepared 1e6 Jurkat cells by washing them twice with PBS and resuspended the cell pellet with 45 μL Nucleofector Solution V (Lonza) and subsequently mixed in 10 μL Supplement (Lonza), 5 μL of 96 μM Alt-R Cas9 Electroporation Enhancer (IDT), 20 μL of crRNA1-containing RNP and 20 μL of crRNA2-containing RNP. We transferred the cells to a cuvette supplied by the Nucleofector reagent, chilled it on ice for 5 min, and performed electroporation with a Nucleofector I device (Lonza) using a preset program (X-001). We let the cells recover in RPMI1640 media supplemented with 10% FBS for several days prior to single-cell clonal selection. We performed isogenic clone selection by allowing cells to undergo gradual expansion from a 96-well plate (seeded with a single cell per well on average) to larger vessels to obtain sufficient cell numbers for subsequent genotype confirmation. We used western blot detection of TMEM16F by an in-house generated rabbit anti-TMEM16F polyclonal antibody (#3016)[15] to screen for clones that do not display TMEM16F antibody cross-reactivity. Next, we isolated genomic DNA from the clones that passed the first round of selection, performed TA cloning, and submitted the miniprep plasmids for Sanger sequencing. Briefly, we added 25 mM NaOH, 0.2 mM EDTA to cell pellets, incubated the samples at 98 °C for an hour, and neutralized with 55 mM Tris-HCl (pH 5.5) to obtain genomic DNA (gDNA). To amplify the targeted genomic fragment for Sanger sequencing, we mixed gDNA with a primer set that recognizes the region outside the pair of guide RNA sequences (forward: AGT TGG TTT CTT GTT CTG CGT T; reverse: ACT CTT GCT CTG GCT TGA TGA T) together with 2x GoTaq master mix (Promega) and commenced the reaction according to the manufacturer's protocol. After confirming the presence of amplicon in the PCR product, we purified the amplicons and

performed TA cloning using a kit (Invitrogen). Plasmids from 8–10 colonies from a single TA cloning reaction were cultured for miniprep and submitted for Sanger sequencing to confirm their genotype prior to further experimentation.

## RT-PCR analysis

We harvested WT and 16FKO Jurkat cells by centrifugation and extracted RNA using an RNeasy mini kit from Qiagen. Next, we prepared cDNA using a ProtoScript First Strand kit (NEB) and used 1 µL of cDNA for 10 µL of PCR using 2x GoTaq master mix (Promega) and 250 nM of forward and reverse primer sets with conditions recommended by the manufacturer. Primer sets were as follows: PIEZO1 forward: GCT GTA CCA GTA CCT GCT GTG; PIEZO1 reverse: CAG CCA GAA CAG GTA TCG GAA G; β-Actin forward: GAC ATG GAA GCC ATC ACA GAC; β-Actin reverse: AGA CCG TTC AGC TGG ATA TTA C. PCR products were analyzed on 2% agarose gel and visualized with the aid of GelGreen nucleic acid gel stain (Biotium).

## Optimization of SLAPSHOT labeling for adherent and suspended cells

U2OS cells on a 3.5 cm dish were washed thrice with PBS in situ without lifting. Jurkat cells were washed twice with PBS, and 1e5 cells were collected for single labeling using centrifugation (300×$g$, 3 min). We added 0.5 mL of solution containing 2× APEX2/BxxT (APExBIO) in PBS to the cells, gently swirled for ~10 s, and initiated the labeling reaction by adding an equal volume of 2× $H_2O_2$ in PBS. After the desired labeling time with constant and gentle mixing, the reaction was quenched by adding an equal volume of 2× ice-cold quencher solution in PBS. The cells were washed twice with the quenching solution before harvesting for proteomics or fixing for microscopy. The labeling reaction parameters were optimized for U2OS and Jurkat cells to the following values, in the following order (all cited concentrations 1×): 0.5 mM BxxT concentration and APEX2 activity (all combinations of 0.125, 0.25, or 0.5 mM BxxT and 0.000125, 0.00025, or 0.0005 AU s$^{-1}$ µL$^{-1}$ APEX2, with 1 mM $H_2O_2$, 1 min labeling time, room temperature); $H_2O_2$ concentration (0.0625, 0.125, 0.25, 0.5, or 1 mM $H_2O_2$, with 0.5 mM BxxT and 0.0005 AU s$^{-1}$ µL$^{-1}$ APEX2); labeling time and temperature (10, 20, 30, 45, or 60 s at room temperature, and 30, 60, 90, and 120 s on ice, with 0.5 mM $H_2O_2$).

## Sulfo-NHS-biotin and cell surface capture labeling

We labeled U2OS cells using SLAPSHOT as described above, sulfo-NHS-biotin (Pierce Thermo Scientific) according to the manufacturer's protocol, and Cell Surface Capture (CSC)[9]. Briefly, sulfo-NHS-biotin labeling was performed by incubating washed cells with 1 mL of 2 mM reagent in HBSS (30 min, 4 °C, slow orbital shaking). After quenching and washing the cells three times with 1 mL of 10 mM Tris-HCl (pH 8.0), 150 mM NaCl, the cells were ready for hypotonic lysis. CSC was performed by incubating washed cells in 1 mL (3.5 cm dishes) or 3 mL (10 cm dishes) of 1.6 mM Na-periodate (NaIO$_4$) in HBSS (20 min, 4 °C, in dark slow orbital shaking). After removing the soluble reagents and washing, the cells were incubated in the same volumes of 10 µM aniline and 10 µM biocytin hydrazide (90 min, 4 °C, in dark, slow orbital shaking). After removing the soluble reagents and washing, the cells were ready for hypotonic lysis or proteomics.

## Cellular fractionation

Cells were hypotonically lysed[60] to obtain crude soluble and membranous fractions after the three labeling methods as well as non-labeled control cells. Briefly, washed cells were scraped in 150 µL of 20 mM HEPES (pH 7.5), 10 mM Mg-acetate, 1 mM EGTA, EDTA-free HALT protease inhibitor cocktail and 10 µg/mL DNase I (>500 Kunitz U/mg, GoldBio) and centrifuged (22,000×$g$, 30 min, 4 °C), to obtain soluble cytoplasmic and insoluble crude membrane fractions. Equal fractions from each fraction were solubilized in 4× LDS sample buffer (Invitrogen) supplemented with 50 mM TCEP and heated (60 °C, 15 min) prior to streptavidin and immunoblotting (see below).

## Visualization of biotin signal with microscopy

We labeled Jurkat cells with SLAPSHOT, sulfo-NHS-biotin, and CSC according to the abovementioned protocols. After washing away the excess reagents (800×$g$, 5 min), we fixed the cells with 2% paraformaldehyde (PFA) in PBS at room temperature for 10 min, quenched with 50 mM ammonium chloride solution in PBS at room temperature for 15 min, performed blocking with 5% BSA in PBS, and incubated cells with streptavidin-conjugated with Alexa Fluor 488 (S11223, Invitrogen) at a concentration of 2 µg/mL for overnight at 4 °C. We performed three washes with PBS between each step and counter-stained with Hoechst 33342 (1 µg/mL in PBS, Invitrogen) in the final step of washing. We mounted the cells with Fluoromount-G (Southern Biotech) on 18 mm coverslips. We acquired the Alexa Fluor 488 and Hoechst 33342 signals on a Leica SP8 microscope with a 63x/NA1.40 objective with the default laser settings for each fluorophore. Images were exported to Fiji software[58] for further compilation.

## Live-or-Dye staining

U2OS cells were grown on 12 mm coverslips and labeled with SLAPSHOT according to the described protocol. After quenching and washing the cells, we added freshly diluted Live-or-Dye 488/515 reagent (Biotium, 1:1000 dilution) in PBS and incubated the cells on ice for 30 min. The cells were washed once with PBS and fixed with 2% PFA in PBS at room temperature for 10 min. We then washed the cells with PBS twice, counter-stained with 1 µg/mL Hoechst 33342 (Invitrogen) in PBS for 5 min and mounted the cells with Fluoromount-G (Southern Biotech). For the negative control, we added Live-or-Dye 488/515 working solution to cells without any manipulations. For the positive control, we treated the cells with 10% ethanol in PBS for 10 min at 37 °C before staining. We acquired the Live-or-Dye 488/515 and Hoechst 33342 signals on a Leica SP8 microscope with a 40x/NA1.30 objective with the default laser settings for each fluorophore. Images were exported to Fiji software[58] for compilation.

## SLAPSHOT and CSC labeling of U2OS cells for proteomics

The U2OS cells were washed thrice and labeled either with CSC or SLAPSHOT in a final volume of 3 mL with constant gentle swirling or left as non-labeled controls. After quenching and washing with ice-cold reagents, we added RIPA buffer without TCEP (see Streptavidin- and immunoblotting) to each plate and incubated it on ice for 15 min. After scraping and collecting the lysates, samples were cleared by centrifugation, and the supernatants were transferred to fresh tubes, snap-frozen, and stored at −80 °C until subsequent NeutrAvidin pull-down and shotgun proteomic experiments. The abovementioned experiments were replicated five times.

## SLAPSHOT labeling of calcium-stimulated Jurkat cells for proteomics

We washed Jurkat cells twice in warm PBS and aliquoted 5e6 cells in 2 mL of PBS containing 1.8 mM CaCl$_2$ into a 15 mL conical tube. We then added ionomycin (Cayman Chemical) to a final concentration of 1 µM and continued incubation at 37 °C for either 1, 5, 10, or 30 min, or left the cells untreated as controls (0 min). At the stimulation endpoint, we added 0.5 mL of ice-cold solution containing 0.005 AU s$^{-1}$ µL$^{-1}$ APEX2, and 5 mM BxxT (all 10×) in PBS, and after a few seconds of gentle mixing, 2.5 mL of ice-cold 1 mM $H_2O_2$ in PBS to initiate the labeling. For each labeling reaction, we also performed mock labeling where the BxxT was omitted. After 90 s of labeling on ice with gentle swirling, we quenched the labeling reaction by adding 5 mL of ice-cold 20 mM Na-azide, and 20 mM Na-ascorbate in PBS. We then washed the cells twice in the ice-cold quenching buffer and once in PBS with centrifugation (500×$g$, 3 min, 4 °C) and snap-froze the well-drained cell pellets. We stored the cell pellets at −80 °C until lysis. The abovementioned experiments were replicated three times.

## Isolation of extracellular vesicles from ionomycin-stimulated Jurkat cells

We stimulated 15e6 WT and 16FKO Jurkat cells with ionomycin in the presence of extracellular calcium as described above, including a non-

stimulated (0 min) control (in triplicate), with appropriately scaled reagent amounts. At the stimulation endpoint, we first removed cells by centrifugation (1000×g, 2 min, 4 °C). Extracellular vesicle-containing supernatant was mixed with EDTA-free HALT protease inhibitor cocktail (ThermoFisher Scientific) and EDTA (5 mM final concentration), and subjected to another round of centrifugation (12,000×g, 30 min, 4 °C) to remove cell debris. We mixed the clarified supernatant with 0.5 volumes of the Exo-spin precipitation buffer (Cell Guidance Systems) and incubated it on rotisserie (18 h, 4 °C). We collected particulate materials containing extracellular vesicles by centrifugation (12,000×g, 90 min, 4 °C). We resuspended the pellets in 100 μL of PBS, loaded the sample onto the Exo-spin columns, and eluted the extracellular vesicles with 180 μL of PBS. We added 4× LDS buffer and heated the samples. We resuspended the cell pellets in RIPA buffer as described below.

### Streptavidin- and immunoblotting

We added RIPA lysis buffer (50 mM Tris (pH 7.4), 150 mM NaCl, 1% NP-40, 0.5% Na-deoxycholate, 0.1% Na-dodecyl sulfate, 10 mM (tris(2-carboxyethyl)phosphine) (TCEP), 1 mM EDTA, EDTA-free HALT protease inhibitor cocktail, 10 mM Na-azide, 10 mM Na-ascorbate) to control and labeled U2OS cells, left them on ice for 15 min, and centrifuged (16,100×g, 15 min, 4 °C). We mixed an equal portion of the lysate supernatant with 4× LDS sample buffer (Invitrogen) supplemented with 50 mM TCEP and heated them at 60 °C for 15 min. We carried out protein gel electrophoresis with Mini-PROTEAN TGX Stain-Free gels (Bio-Rad) and applied UV light (5 min, ChemiDoc Touch imaging system, Bio-Rad) to activate the stain-free labeling. We then transferred the proteins to Immobilon-P membranes (EMD Millipore) with the Trans-Blot transfer system (Bio-Rad) using 25 mM Tris, 192 mM glycine, and 20% (v/v) ethanol transfer buffer. We then quantified the stain-free labeled total protein on the transferred membranes using the ChemiDoc Touch imaging system. We detected the biotin incorporation using streptavidin-HRP conjugate (#016-030-084, Jackson ImmunoResearch) after blocking with 3% BSA dissolved in TBST (10 mM Tris-HCl (pH 8.0), 150 mM NaCl, 0.1% Tween 20). For TMEM16F immunoblotting, we blocked the membrane with 5% non-fat milk dissolved in TBST, incubated with in-house generated anti-TMEM16F primary antibody[15] (1:2000 dilution), and donkey anti-rabbit IgG-HRP secondary antibody (#711-035-152, Jackson ImmunoResearch, 1:24,000 dilution). To detect GAPDH, we used an anti-GAPDH-HRP antibody (#HRP-60004, Proteintech Group, 1:60,000 dilution). We washed the membrane extensively with deionized water, followed by TBST after each antibody incubation. We used a C-DiGit scanner (LI-COR) for chemiluminescence detection after incubating with either Pierce ECL (for streptavidin and GAPDH blots, Thermo Scientific) or SuperSignal West Pico Plus substrate (for TMEM16F blot, Thermo Scientific). To analyze extracellular vesicle preparation or the remaining cell pellets after ionomycin stimulation, we loaded equal volume of protein samples onto Bolt Bis-Tris plus mini protein gels (Invitrogen) resolving with NuPAGE MOPS SDS running buffer (Invitrogen) and transferred the proteins to nitrocellulose membranes (Bio-Rad) with a Trans-Blot transfer system (Bio-Rad) using NuPAGE transfer buffer containing 20% (v/v) ethanol. Membranes were handled as described above with the following variations: anti-TMEM16F antibody (1:1000 dilution), ACSL1 (#13989-1-AP, ProteinTech Group, 1:1000 dilution), α-tubulin-HRP (#HRP-66031, ProteinTech Group, 1:25,000 dilution), DGAT2 (#17100-1-AP, ProteinTech Group, 1:1000 dilution), and anti-GAPDH-HRP antibody (1:25,000 dilution). Blots incubated with non-HRP conjugated antibodies were further exposed to Goat anti-rabbit IgG-HRP secondary antibody (#711-035-152, Jackson ImmunoResearch, 1:72,000 dilution). All incubations with primary antibodies (including HRP conjugates) were carried out overnight at 4 °C and secondary antibodies for at least 1 h at room temperature. Chemiluminescence substrates (all from Thermo Scientific) were used as follows: Pierce ECL (α-tubulin and GAPDH blots for cell pellets), SuperSignal West Pico Plus (α-tubulin and GAPDH blots for extracellular vesicles, ACSL1 and DGAT2 blots for cell pellets), SuperSignal West DURA (TMEM16F blots for cell pellets), and

SuperSignal West Femto (ACSL1, DGAT2, and TMEM16F blots for extracellular vesicles).

### Densitometric analysis of western blots

Background-corrected protein band intensities were obtained using the Average background analysis tool in Image Studio Lite software (LI-COR), with border width set at "1" for all segments. The normalized densitometric values were obtained by equalizing the means of the band intensities across the triplicate assays. Additionally, for plotting purposes, data were scaled so that the mean of the WT samples equals one.

### Capture and processing of biotinylated proteins

We incubated cleared cell lysates prepared in RIPA buffer without TCEP (see Streptavidin- and immunoblotting) with (slurry volume) 125 μL (Jurkat) or 50 μL (U2OS) of pre-washed high-capacity NeutrAvidin-agarose beads (ThermoFisher) on the rotisserie for 30 min at 4 °C (all sample handling from this point on was performed at 4 °C or on ice until otherwise noted) and removed the unbound material by centrifugation (1000×g, 5 min, 4 °C). One percent aliquots of the lysates and the unbound fractions were saved for streptavidin blot analysis to ensure proper titer of beads and quantitative capture of biotinylated proteins. The beads were washed five times with the lysis buffer without quenchers (additionally, protease inhibitors were omitted starting from the second wash), five times with 1 M NaCl, 50 mM Tris-HCl (pH 7.4) buffer, and five times with 1.5 M urea, 250 mM ammonium bicarbonate (digestion buffer). Each wash was performed using 1 mL of buffer for 5 min (4 °C) on rotisserie, followed by centrifugation. We then reduced and alkylated the samples with 250 μL of 10 mM Tris(2-carboxyethyl)phosphine) (TCEP), and 40 mM iodoacetamide (IAA) in the digestion buffer for 30 min at room temperature protected from light. After removing the soluble reagents, we digested the samples on-bead in 250 μL of the digestion buffer containing 0.1% (w/v) RapiGest SF surfactant (Waters), 1 mM CaCl₂, and 1 μg of trypsin/Lys-C (Promega) on a rotisserie for 18 h at ambient temperature. After digestion, we collected the soluble materials and washed the beads twice with 250 μL of 10 mM ammonium bicarbonate and pooled all unbound materials.

We acidified the digested samples with concentrated trifluoroacetic acid (TFA) and cleaned up the peptides with C18 SOLA Solid-Phase Extraction (SPE) columns (ThermoFisher). Briefly, we re-loaded the flow-through from the first loading, washed the column first with 1 mL of 0.1% TFA and then with 0.1% formic acid (FA), and eluted bound materials with 0.1% FA in 60% acetonitrile. Loading and elution steps were performed unaided (by gravity), whereas the washes were aided by gently applying air pressure via a syringe. The eluted materials were dried in vacuum, redissolved in water, and quantified spectrophotometrically by A₂₈₀ measurement on NanoDrop (ThermoFisher) using BSA peptide mixture as a standard. For obtaining fair and unbiased intensity information between samples from a given experiment, an equal fraction of the total peptide material from each sample in the experiment was used for injection or isobaric labeling.

### Isobaric labeling of peptides

We utilized an optimized protocol[61] for tryptic peptide labeling with a TMT10plex Isobaric Labeling Reagent kit (Thermo Scientific). Unlabeled (control) and labeled samples from the five time points of a single replicate from a given cell type comprised a set that was labeled with the ten individual reagents. Specifically, the non-SLAPSHOT-labeled controls with 0, 1, 5, 10, and 30 min calcium stimulations were correspondingly labeled with the 126, 127N, 127C, 128N and 128C reagents, and the SLAPSHOT-labeled samples at the same time points with the 129N, 129C, 130N, 130C and 131 reagents. A total of six sets of samples were labeled, corresponding to each of the triplicate experiments for the WT and 16FKO cells. After quenching the reaction with hydroxylamine, the samples from a given set were pooled, dried, redissolved, and fractionated with a Pierce High-pH Reversed-Phase Peptide Fractionation Kit (Thermo Scientific) into eight fractions according to the manufacturer's instructions. We measured the peptide concentration

from each dried fraction by NanoDrop. The resulting peptide mixtures were analyzed by LC-MS/MS.

## Analysis of the amino acid targets of the phenoxyl radicals

We harvested AMO1 cells by centrifugation (500×*g*, 3 min) and washed twice in PBS. The cell pellet was dissolved in 7.5 M urea 250 mM ammonium bicarbonate buffer (~10e7 cells/mL) with sonication. After clarification (16,100×*g*, 15 min, 4 °C) and measuring protein concentration by BCA assay, the proteins were reduced and alkylated with TCEP and IAM. After diluting the sample with 250 mM ammonium bicarbonate to a final urea concentration of 1.5 M, $CaCl_2$ was added to a final concentration of 1 mM, trypsin/Lys-C (Promega) and 0.1% (w/v) RapiGest SF surfactant (Waters) and trypsin/Lys-C to 1:50 protease:protein ratio. The digestion was allowed to proceed for 18 h at ambient temperature. We cleaned the digested peptides by SPE and quantified them by NanoDrop. For the analysis of the labeling targets of APEX2, we prepared two vials, each with 5 µg of the cleaned-up peptides, 0.001 AU s$^{-1}$ µL$^{-1}$ APEX2, and 1 mM BT in a total volume of 5 µL in PBS. APEX2 labeling was then initiated in one of the vials with 5 µL of 1 mM $H_2O_2$ in PBS, whereas the other was left as control receiving PBS. After 45 s incubation, 10 µL of quenching buffer with 20 mM Na-azide, and 20 mM Na-ascorbate in PBS was added to both vials. We applied the reaction mixtures to a 3000 MWCO spin filter (NanoSep, Pall) to remove APEX2 and the formed colloid, and depleted the remaining biotinylated materials from the filtrate with 25 µL (slurry volume) of Pierce streptavidin magnetic beads (Thermo Scientific). Peptides in the supernatants were cleaned up with C18 ZipTips (EMD Millipore) prior to LC-MS/MS analysis.

## Mass spectrometry

We performed LC-MS/MS analyses with data-dependent acquisition (DDA) on a Thermo Scientific Q Exactive Plus Hybrid Quadrupole-Orbitrap mass spectrometer interfaced to a Dionex UltiMate 3000 UHPLC system with a NanoSpray Flex source. The MS instrument was operated in the positive mode with spray voltage and heating capillary temperature set to 2200 V and 250 °C, respectively. We injected ~1 µg of the peptide material in 5 µL of aqueous 0.1% FA (Solvent A) into an Acclaim PepMap RSLC C18 column (75 µm × 150 mm, 2 µm particle size, 100 Å pore size, Thermo Scientific) kept at 40 °C. Chromatographic separations were achieved by an increase in 0.08% FA in 80% acetonitrile (Solvent B) with the following gradient: 0 to 15 min: 3% B at 500 nL/min; 15 to 210 min: 3 to 50% B at 200 nL/min, followed by column washes at 500 nL/min.

Survey (MS$^1$) DDA scans were recorded with 350–1500 *m/z* (label-free samples) or 375–1400 *m/z* (TMT-labeled samples) scan range, 70,000 resolution (at 200 *m/z*), 1.7 *m/z* (label-free) or 0.7 *m/z* (TMT) isolation window, 3e6 AGC target, 100 ms (label-free) or 50 ms (TMT) maximum IT, 20 s (label-free) or 30 s (TMT) dynamic exclusion, exclusion of isotopes, as well as unassigned and +1 charge states, preferred peptide matches, and 27 (label-free) or 32 (TMT) NCE. A maximum of 12 (label-free) or 15 (TMT) HCD fragment (MS$^2$) scans were acquired with fixed mass 100 *m/z*, 17,500 (label-free) or 35,000 (TMT) resolution (at 200 *m/z*), 5e4 (label-free) or 1e5 (TMT) AGC target, 2e3 minimum AGC target, and 180 ms (label-free) or 100 ms (TMT) maximum IT. We saved all spectra in the profile mode.

## Proteomics data analysis

We analyzed the RAW data files containing LC-MS/MS spectra with MS$^1$-quantitation-based MaxQuant[62] (v1.6.17.0) against the human proteome FASTA file containing the reviewed canonical and isoform entries (obtained from UniProt on 2020-04-02) as well as the His-APEX2 sequence. We used the following search parameters: Trypsin/P digestion, 2 max missed cleavages, oxidation (M) and acetyl (protein N-term) as variable modifications and carbamidomethyl (C) as fixed modification. All other settings were left with their default values, including a 1% false discovery rate (FDR) at both peptide and protein levels. We invoked a match between runs for the TMT and peptide depletion experiments and Intensity-Based Absolute Quantification (iBAQ) for the method development and TMT and experiments.

Reverse hits, as well as potential contaminants and protein groups only identified by site, were removed from consideration, as were proteins that were identified but had zero intensity in all samples in the experiment. For the experiments using U2OS cells comparing SLAPSHOT to CSC, we subtracted the Intensity-Based Absolute Quantitation (iBAQ) values of the controls from those of the two labeled samples to obtain intensities free of contributions from the non-specifically bound materials to the solid support. For the experiments using Jurkat cells, to obtain protein intensities that can be compared both within and among samples, we implemented the relative iBAQ-TMT (riBAQ-TMT) quantitation scheme[63], whereby the TMT channel intensities for a given protein were scaled so that the sum from all channels equals the iBAQ value[64]. We then subtracted the intensities of non-labeled control channels from those of the corresponding labeled channels. Proteins with resulting zero or negative intensity were considered not detected in the sample. For the experiments on Jurkat cells, in the statistical R computing environment[65] (v4.0.5 or newer), we applied miceRanger[66] (v1.3.4) to impute missing values and internal reference scaling (IRS) normalization with pseudo-reference[67] to correct for batch effects. The imputation was however ignored, and protein intensities left to zero if the protein was not detected in any of the six samples (three labeled and three controls) from a given condition.

## Bioinformatics analyses

Differential protein expression (fold-change and *p*-value) for log$_2$-transformed iBAQ or riBAQ data was assessed by paired two-tailed Welch's tests in Perseus[68] (v1.6.15.0) without multiple test adjustments. We applied absolute log$_2$ fold-change >1, and/or *p*-value < 0.05 as criteria for statistical significance where applicable. We utilized protein subcellular localization annotations and other information from UniProt[69], Human Protein Atlas[70], (v19.3, *proteinatlas.org*), VesiclePedia[71] (v4.1), and Surface Protein Atlas[28] (see Supplementary Data 7 for the compiled data from these sources). Pathway and gene set analyses were performed in Protein ANalysis THrough Evolutionary Relationships (PANTHER)[72,73] (v16), and Reactome[74] (v79). PANTHER analyses were performed with Fisher's exact test without multiple test adjustments. Redundant and overlapping categories were removed. Search Tool for Retrieval of Interacting Genes/Proteins (STRING) analysis[75] (v11.5) in the Cytoscape environment[76] (v. 3.9.1) was used to derive and visualize physical associations between proteins. A diverse set of theoretical model profiles (N = 2182) were constructed (Supplementary Data 5) that vary both in the directions and magnitudes of intensity changes between the time points. The initial model set was filtered to contain only those models (N = 2109) with at least 50% difference between the smallest and largest value, in addition to the "static" model. Protein intensity profile distances to each model profile were calculated in R with the philentropy package[77] (v. 0.6.0) using Bhattacharyya distance, and the protein was assigned a profile with the model with the smallest distance. The descriptions were simplified by collapsing the models with the same general pattern, regardless of the intensity information, into one of the simplified 'x-x-x-x' type models (N = 61, plus "not detected") where each 'x' informs of the direction of the intensity change between two consecutive time points. Each 'x' can be either 'd' (for down), 'c' (for constant), or 'u' (for up). The models do not cover every possible scenario, of which there are $3^4 = 81$, as some a priori rare or less plausible patterns were omitted. Enrichment factor (E) of the overlapping proteins belonging to particular profiles in the WT and 16FKO cells was calculated with E = $(n \times N) / (wt \times ko)$, where *n* is the number of overlapping proteins, *N* is the total number of detected proteins (4148), wt is the number of proteins with a particular profile in the WT cells, and ko is the number of proteins with a particular profile in the 16FKO cells. The *p*-values for the overrepresentation and underrepresentation of the overlaps were calculated with Fisher's exact test in base R. Hierarchical clustering for the log$_2$-transformed data was performed in R with the ComplexHeatmap package[78] (v1.10.2) with method and metric set to single and Pearson correlation, respectively.

## Statistics and reproducibility

Quantitative experiments, including proteomics and western blotting, were performed at least in triplicate using separate cell passages for each experiment. Sample sizes were dictated by feasibility as well as limitations in mass spectrometer time and other resources rather than power calculations. Multiple hypothesis testing corrections were omitted as our interests were mainly in groups of proteins rather than individual proteins. Unless otherwise specified, statistical data are reported as mean ± standard deviation. All $t$-tests performed were of the Welch's variety by default. Where applicable, group differences were considered significant when $p$-value < 0.05.

## Reporting summary

Further information on research design is available in the Nature Portfolio Reporting Summary linked to this article.

## Data availability

The mass spectrometry proteomics data have been deposited to the ProteomeXchange Consortium (http://proteomecentral.proteomexchange.org) via the PRIDE partner repository[79] with the dataset identifier PXD041387. All other data that support the findings of this study are included within this manuscript and its Supplementary Information files. Supplementary Data and supporting data for figures are presented in the Supplementary Data 1–7 file. Uncropped blots for Supplementary Fig. 18a are provided in Supplementary Fig. 18c.

## Materials availability

The plasmid encoding the soluble hexahistidine-APEX2 is available at Addgene with ID #201471.

## Code availability

All R code used to generate data in this paper is presented in Supplementary Data 8.

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

## Acknowledgements

We thank Drs. Wen Lu and Arthur Weiss for providing Jurkat E6.1 cells as well as their insight into Jurkat cell biology, Dr. Christoph Driesen for providing AMO1 cells, Dr. Alice Ting for providing APEX2-NLS plasmid, Drs. Mark Burlingame and Kamal Mandal for assistance with mass spectrometry, Drs. Ya-Chu Chang, David Crottès, John Gilchrist, and Han-Hsuan Liu for helpful suggestions and critical readings of the manuscript, and Jan and Wiita lab members for their input. C.F.T. is a recipient of a UCSF PBBR post-doctoral grant that sponsored part of this work. This study is supported by R35 NS122110 to L.Y.J. and DP2 OD022552 to A.P.W. L.Y.J. and Y.N.J. are investigators of the Howard Hughes Medical Institute.

## Author contributions

S.T.T. and C.F.T. conceived the project, executed all experiments, analyzed and interpreted the data, and wrote the manuscript. Y.N.J., L.Y.J., and A.P.W. oversaw the project and provided the resources for the project. All authors discussed and commented on the results and participated in editing the manuscript.

## Competing interests

The authors declare no competing interests.
