## [Peer Review File · Communications Biology]

Reviewer #1 (Remarks to the Author):

In their manuscript, Tuomivaara et al report on SLAPSHOT, a new proximity proteomics method to identify the extracellularly exposed proteome in live cells. This technique relies on modification of the APEX method to allow for investigation of endogenous proteins, with no requirement for cell or protein engineering. After demonstrating optimization efforts for the use of SLAPSHOT in cells, the authors apply this strategy to understand dynamic changes in the extracellularly exposed proteome of Jurkat cells in response to CRISPR-mediated deletion of the calcium-dependent phospholipid scramblase and ion channel protein TMEM16F. These studies demonstrated differences in proteins identified based on TMEM16F expression, with WT cells displaying more proteins from the ER and KO cells displaying more proteins from the mitochondria. Although the authors have identified an important need for technologies that facilitate labelling of the cell surface proteome to answer important questions like the function of TMEM16F in regulating it, I have identified several concerns with this study in its current form:

- 1) The manuscript is very long and would benefit from more focused, concise reporting of background information, the results and discussion. In particular, the discussion was abnormally lengthy and seemed to re-iterate the findings more than position the results within the field. That being said, the authors did well in referencing the current literature.
- 2) There are blanks/highlighted areas where data deposition information should have been input. This is essential for an open peer review process.
- 3) I also was surprised that no table was included showing the individual intensity data for each gene identified across the WT and KO replicates. This is usually provided with the paper as an excel file. The file that was included seems to compare the 5 replicates that were used for the initial optimization of the SLAPSHOT method. However, even here the authors only report on 'detected' or not. This doesn't give the reader any information on the intensity of the detection or what the consistency across runs is like. There is no table included that compares identifications for WT vs TMEM16F KO.
- 4) From what I gather, the authors used MS1 quantification of DDA data, although this was not implicitly stated. This is still the most reported quantification method in the field, however there is a movement towards DIA quantification methods, that are less stochastic than those that involve selection of a limited number of precursor peptides for fragmentation. At minimum I would comment on the limitations of this 'quantification' method.
- 5) The authors did a good job of positioning the unique utility of their technique in comparison to the wave of similar methods that have recently been reported on. While PECSL (Li et al., 2021) is a similar strategy using HRP, other APEX-based strategies involve

protein engineering to anchor the enzyme to the cell surface (Li et al., 2020; Kirkemo et al., 2022). The uniqueness of their strategy would seem to shine most in its use for systems where protein engineering is not possible, such as in tissues; however, they chose to use it in cell systems, where I think there is less novelty in the method, and they are unable to demonstrate a clear advancement using their method. I would also like to note that the authors indicated APEX is ideal versus other strategies because it targets Tyrosine and Tryptophans and not Lysines, and that targeting lysines can interfere with proteolysis at later steps of processing. Although theoretically logical and there are some studies reporting this to be the case, there is rather weak evidence of this actually negatively impacting identifications in practice, and I do not think it should be highlighted as a major advantage of the SLAPSHOT method.

6) The authors compared their method to the classical CSC method and while they captured significantly more proteins in significantly less time, the novel proteins identified were mostly enriched for those that are not typically described on the cell surface. This suggests that the methodology may require further development rather than being superior as the authors present it. Although there was not evidence of labelling in the interior of fixed cells, there remains the possibility that labelling is occurring post lysis using this method, leading to a large increase of intracellular protein labeling. Additional controls may assist in teasing this out, but as it stands, I am not confident that this method truly labels only the extracellular exposed proteome in live cells. The authors surmise that some of these identifications could be related to exosomes from the ER or mitochondria, but based on the methods they've described, where successive washing is performed prior to cell lysis, I wouldn't expect exosomes to have been collected and thereby identified by MS.

7) The authors present the timecourse data not as intensity values for each protein identified, but rather based on its identification as being increased (u), decreased (d) or constant (c) as compared to the intensity value in the previous timepoint. Although these comparisons are important, I found this binning method needlessly confusing. And when you actually look at the intensity values for the select proteins that they did present in their more raw form (SF17), they did not seem to match the binning value at all. For example, from the heatmap in figure 4, the intensities observed across the d-u-u-u subgroup seem to have very similar intensities across the groups, and I would have expected them to be labelled c-c-c-c instead? I could not actually check the intensity values myself because they were not provided to me (see comment 3). In SF17, they reported on a few of the proteins in each of these groups, but the classes they reported did not seem to visually match up with the intensities at all. Eg: LFA-1, 0-1min:WT says down, but if anything it is slightly up? More likely constant. This is repeated throughout the figures. The authors also report on changes observed between incredibly low intensities (<1000), which I find very problematic to have confidence in.

8) In Figure 5B, the Y axis has been cut off. Both TMEM16F and PIEZO1 have very low intensity values in WT, so I would not say it's fair to report those proteins as 'missing' from the extracellular exposed proteome because they are hardly detected in the WT samples.

9) In the discussion of the results, the authors provided a lot of supposition about what their findings could suggest about TMEM16F functions. However, they did not test any of these hypotheses. Indeed as with most omics studies, there are a lot of possible biological changes that can be followed-up on, and not all can be carried out, but by not following up on any of the hypotheses generated by the author's interpretation of the results, I find it challenging to have confidence in the method itself being able to successfully report on new biology. The sheer number of specific interpretations of the results without evidence is very problematic to me, especially when there is concern about the accuracy of the technique itself.

Reviewer #2 (Remarks to the Author):

The authors present a novel enzymatic labeling method for proteomic analysis of the surfaceome, alternative to Cell Surface Capture or lysine biotinylation using sulfo-NHS-biotin. The presented benefit of this method lies in the possibility of tracking temporal dynamics of the proteome thanks to the labeling efficiency at low temperatures. The article contains details of the optimization procedure and a comparison with the established methods for surface labeling. The core of this work lies in the comparative quantitative time-course proteomic analysis of wild-type cells and cells lacking TMEM16F in response to calcium stimulation, performed using the SLAPSHOT method. The method is well suited for the biological issue under scrutiny and could be employed in a broad spectrum of biological issues.

The article is robust yet clear overall, and the amount of work done is impressive. Yet, I have several specific recommendations and also a few doubts, which I believe may be refuted by the authors:

1. Some changes that should be made in the Introduction and beginning of the Results for more clarity:

Comparison of the SLAPSHOT method with the established surface labeling approaches is made in the results. The discussion then contains arguments further comparing these methods. The statement that this method presents an alternative to these methods should also be mentioned in the introduction.

The introduction omits information about which molecule specifically is used for enzymatic labeling. This information first appears in the results section, even the fact that the method uses biotinylation first appears in the results. Similarly, the fact that the

method uses the label BxxT is mentioned (p. 5) before it is actually explained (p. 6) what this molecule is and why it is used.

2. Figure 1B is problematic.

The data presented in Figure 1B, the part stained with Streptavidin-HRP, are not informative, as the signal intensities vary among the different methods. The differences between soluble fraction and pellet are incomparable, as some “S” lanes are likely below the detection limit. The figure would be informative if parts of the blot for each of the methods were detected separately and the intensity of the “P” lanes visually at least approximately matched.

Moreover, this figure is supposed to demonstrate that the biotinylation in the SLAPSHOT method is mostly restricted to the membrane fraction. However, I am not sure I can assess if a correct approach was made since the related section in the Methods (“Other labeling methods and hypotonic crude fractionation,” p. 22) cites “Teo and Wells, 2014”, which does not contain any information about the method used for hypotonic cell lysis, or fractionation leading to isolation of membrane fraction, nor does any other section of the Methods.

3. The explanation of the presence of intracellular soluble proteins in the SLAPSHOT analysis.

The comparison of SLAPSHOT and CSC (p. 6 and p. 7) shows that CSC is somewhat more specific in identifying plasma membrane proteins, while SLAPSHOT identifies abundance of intracellular soluble proteins. This may point either to a flaw in the method’s design or to the explanation about the secretome presented by the authors. The microscopic data (p.7-8, figure S6) point to the integrity of the membrane, therefore, label permeability and membrane rupture should not be the issue. However, the presence of His-APEX2 in the washed samples (p. 15, figure S16) leads me to a suspicion that labeling may continue during the capture procedure. Can you rule this suspicion out?

4. Data support for the secretome explanation.

“Notably, these 1679 proteins with intracellular annotation include 1509 (89.9%) that have been detected in the secretome and/or the extracellular vesicle proteome of U2OS cells” (p. 7) – as this is a crucial argument for explaining why are these proteins identified in such abundance when using SLAPSHOT, the overlap should be shown, e.g. in an additional supplementary table. Does CSC detect any of such proteins?

5. Limitations of the method may be discussed in the Discussion section.

For example, “low availability of the phenoxyl radical targets in the proteome” (p. 7) seems like a potential limitation. However, even though it is mentioned in the discussion (p. 13), it is not presented as a limitation.

Another example may be “our initial MS analysis detected ... unique set of proteins with plasma membrane annotation that were not detected by CSC” (p. 17) – there were 3 times more unique plasma membrane proteins detected by CSC. This seems like a limitation

worth mentioning.

6. A potential inconsistency of the text in Results with the data presented in Figure 4B.

“(f) Curiously the c-c-u-u pattern in 16F KO cells only contains five proteins but collectively accounts for higher riBAQ value than many other patterns with more protein constituents (Figure 6E).” (p.13) – Figure 4B displays the riBAQ values using the color gradient, but it is not visible that the c-c-u-u pattern in 16F KO cells has any high values. Therefore, it is unclear to me whether this claim is wrongly formulated or mistaken.

7. A dubious statement in the Discussion.

“... Such phenomenon further confirms that mitovesicle release is insensitive to acute intracellular calcium elevation ...” (p. 13) – the increase in mitovesicles happens in both cell lines subjected to Ca influx. Yes, they do not react immediately, but on what do you base the claim that they are insensitive to Ca influx? Additionally, I would not call the different temporal pattern “drastically different” (p. 17)

8. Further minor details.

Introduction:

(p. 3) The abbreviation PECSL is not explained.

Results:

(p. 5) “(see Materials and Methods)” – the reference should be more specific

(p. 7) Supplementary figure A and B – in all cases in the text (figure description and p. 14), A refers to B in the figure and vice versa

(p. 10) ““488 and 460 membrane proteins” – how are membrane proteins defined here, are they based on an annotation or prediction? Similarly, when mentioning annotations (p. 7), it should be stated, which precise annotations from which database were used

(p. 12) “(Figure 5C)” – should be 6C.

(p. 12) “...is the second most populated...” – is actually third, when counting the c-c-c-c pattern

Figures:

(p. 35) The bitmap in 5A covers the top of the graphs in 5B. Also, there is an object in the bottom right corner of the bitmap in 5A that should be cropped out

To whom it may concern:

General remarks to both reviewers:

We sincerely thank both reviewers for their thoughtful and valuable criticisms. We have made adjustments and corrections to the manuscript accordingly and also addressed the issues raised in an itemized table as shown below. As a general note, we have condensed the manuscript to adhere to the journal guidelines. Furthermore, we have added several more supplementary tables, also to adhere to the journal guidelines, to show the underlying data for each figure. Finally, we have modified Supplementary Figure 13 to properly illustrate the data in more detail.

Reviewer #1 (Remarks to the Author):

In their manuscript, Tuomivaara et al report on SLAPSHOT, a new proximity proteomics method to identify the extracellularly exposed proteome in live cells. This technique relies on modification of the APEX method to allow for investigation of endogenous proteins, with no requirement for cell or protein engineering. After demonstrating optimization efforts for the use of SLAPSHOT in cells, the authors apply this strategy to understand dynamic changes in the extracellularly exposed proteome of Jurkat cells in response to CRISPR-mediated deletion of the calcium-dependent phospholipid scramblase and ion channel protein TMEM16F. These studies demonstrated differences in proteins identified based on TMEM16F expression, with WT cells displaying more proteins from the ER and KO cells displaying more proteins from the mitochondria. Although the authors have identified an important need for technologies that facilitate labelling of the cell surface proteome to answer important questions like the function of TMEM16F in regulating it, I have identified several concerns with this study in its current form:

	Comments	Responses
1	The manuscript is very long and would benefit from more focused, concise reporting of background information, the results and discussion. In particular, the discussion was abnormally lengthy and seemed to re-iterate the findings more than position the results within the field. That being said, the authors did well in referencing the current literature.	We appreciate the suggestions and the compliment. We have condensed all parts of the manuscript significantly for the readers' convenience and to adhere to the journal word limit. We truncated the word count from 6088 to 4501.
2	There are blanks/highlighted areas where data deposition information should have been input. This is essential for an open peer review process.	We apologize for leaving out the data deposition details. The situation has now been corrected (lines 772 and 776). The reviewers can access the proteomics data at the PRIDE repository with dataset identifier PXD041387 with username: reviewer_pxd041387@ebi.ac.uk and password: 7gc1QBUD. The plasmid ID at Addgene is 201471. The plasmid and MS

		data will be released for public access upon publication of this manuscript.
3	I also was surprised that no table was included showing the individual intensity data for each gene identified across the WT and KO replicates. This is usually provided with the paper as an excel file. The file that was included seems to compare the 5 replicates that were used for the initial optimization of the SLAPSHOT method. However, even here the authors only report on ‘detected’ or not. This doesn’t give the reader any information on the intensity of the detection or what the consistency across runs is like. There is no table included that compares identifications for WT vs TMEM16F KO.	We agree that this information is valuable and should be reported. Tables with intensity information are included for both the method development part with U2OS cells (appended in Supplementary Table 1, as well as the calcium stimulation experiment with Jurkat cells (new Supplementary Table 3). The tables include intensity information for all replicates.
4	From what I gather, the authors used MS1 quantification of DDA data, although this was not implicitly stated. This is still the most reported quantification method in the field, however there is a movement towards DIA quantification methods, that are less stochastic than those that involve selection of a limited number of precursor peptides for fragmentation. At minimum I would comment on the limitations of this ‘quantification’ method.	A comment regarding the limitations of DDA experiments has been added to the Discussion (line 399).
5	The authors did a good job of positioning the unique utility of their technique in comparison to the wave of similar methods that have recently been reported on. While PECSL (Li et al., 2021) is a similar strategy using HRP, other APEX-based strategies involve protein engineering to anchor the enzyme to the cell surface (Li et al., 2020; Kirkemo et al., 2022). The uniqueness of their strategy would seem to shine most in its use for systems where protein engineering is not possible, such as in tissues; however, they chose to use it in	We thank the reviewer’s kind words. We agree that SLAPSHOT would be a useful approach for tissue or other samples that are recalcitrant for genetic manipulation (as mentioned in the discussion). At the inception of this work, we identified two aspects in the extracellularly exposed proteome mapping efforts that SLAPSHOT could have advantage at: (1) the temporal advantage where fleeting changes during fast signaling events can be captured; (2) samples whose genetic manipulation is nontrivial. After careful consideration, we decided to focus on showcasing the

	cell systems, where I think there is less novelty in the method, and they are unable to demonstrate a clear advancement using their method. I would also like to note that the authors indicated APEX is ideal versus other strategies because it targets Tyrosine and Tryptophans and not Lysines, and that targeting lysines can interfere with proteolysis at later steps of processing. Although theoretically logical and there are some studies reporting this to be the case, there is rather weak evidence of this actually negatively impacting identifications in practice, and I do not think it should be highlighted as a major advantage of the SLAPSHOT method. temporal advantage, since there are many clever approaches and technological advancements in addressing the second aspect. That said, we look forward to seeing SLAPSHOT in tissue labeling. We have toned down the section where we discussed the biotinylated lysine effect on trypsin digestion (line 306).
6 The authors compared their method to the classical CSC method and while they captured significantly more proteins in significantly less time, the novel proteins identified were mostly enriched for those that are not typically described on the cell surface. This suggests that the methodology may require further development rather than being superior as the authors present it. Although there was not evidence of labelling in the interior of fixed cells, there remains the possibility that labelling is occurring post lysis using this method, leading to a large increase of intracellular protein labeling. Additional controls may assist in teasing this out, but as it stands, I am not confident that this method truly labels only the extracellular exposed proteome in live cells. The authors surmise that some of these identifications could be related to exosomes from the ER or mitochondria, but based on the methods they've described, where successive washing is performed prior to cell lysis, I wouldn't expect exosomes to have been collected and thereby identified by MS.	We recognize the reviewer's concerns regarding the labeling of canonical intracellular proteins with SLAPSHOT, for we had the same thoughts upon inspecting the proteomics data. As pointed out by the reviewer, microscopic and western blot evidence supports the spatial specificity of SLAPSHOT. After experimenting and testing various possibilities in order to get to the bottom of the matter (as shown in the supplementary materials), we utilized Occam's razor and concluded that these canonical intracellular proteins are indeed exposed to the extracellular milieu. The strength of the classical CSC in surfaceome mapping is that its chemical reaction targets the complex glycans on the plasma membrane and many canonical extracellularly secreted proteins. That said, in our dataset, CSC method also identified more than 1000 canonical intracellular proteins, likely due to the presence of O-GlcNAcylation (an abundant intracellular glycosylation) and carbonylation (another naturally occurred but underappreciated PTM that is a directed target for hydrazide-biotin) proteins. Unlike complex glycosylation, O-GlcNAcylation and

carbonylation display low stoichiometry. Thus, these intracellular proteins are often considered noise and have never been seriously considered in most surfaceome mapping efforts. Also, depending on the scientific question (e.g. if one is only interested on plasma membrane surface biomarker), they are often simply excluded from the discussion.

SLAPSHOT targets polypeptide directly (tyrosine and tryptophan side chains) rather than PTMs. Thus, it does not discriminate the origin of the proteins, as long as they are extracellularly accessible.

We agree with the reviewer that vesicles are likely washed away during the sample processing. Our hypothesis is that many of the “released vesicle proteins” are *en route* to the extracellular space before they have a chance to fully separate from the cells and are labeled and captured due to the fast kinetic afforded by SLAPSHOT. The observation of such phenomenon in vesicle shedding can be found in the videos in several publications, such as PMID: 30622179; 31995754.

We also would like to emphasize that we do not claim that SLAPSHOT labeling is specific for surfaceome (which often equates to plasma membrane residents). Rather, we present SLAPSHOT as a method to capture extracellularly exposed proteins. Additionally, we are not claiming that SLAPSHOT is superior to CSC (although its time resolution is). Rather, we stated that SLAPSHOT could be applied to address certain biological questions that might not be answerable using other methods. We believe all methods have their unique strengths along with shortcomings. Thus, one should consider all aspects of the scientific question at hand and carefully

		choose an appropriate tool from the “toolbox”. We appreciate the comment regarding labeling after the intended labeling period, and this is one of the issues we wanted to rigorously rule out starting from the initial sketching of the experiments. Our experimental design makes post-lysis labeling very unlikely since: (1) The labeling reaction was quenched with known and potent inhibitors (as shown in the supplementary materials as well as the Ting group in their publications) immediately after the labeling period, (2) The cells were washed prior to lysis, removing all water-soluble reagents, also with the quenchers included in the wash buffers, neutralizing any residual reactive enzyme molecules. (3) Lysis buffer itself contains the same quenchers. We also note that sodium azide is an irreversible heme-specific inhibitor, so it will effectively inhibit APEX2 upon contact.
7	The authors present the timecourse data not as intensity values for each protein identified, but rather based on its identification as being increased (u), decreased (d) or constant (c) as compared to the intensity value in the previous timepoint. Although these comparisons are important, I found this binning method needlessly confusing. And when you actually look at the intensity values for the select proteins that they did present in their more raw form (SF17), they did not see to match the binning value at all. For example, from the heatmap in figure 4, the intensities observed across the d-u-u-u subgroup seem to have very similar intensities across the groups, and I would have expected them to be labelled c-c-c-c instead? I could not actually check the	We thank the reviewer for inspecting our data with a fine comb. Figure 4 was wrongly assembled and has now been corrected. The intensities in the heatmap are still somewhat subtle, as most proteins have the c-c-c-c pattern, blunting the visual changes relative to the total intensity. We also included the intensity pattern for each protein in a new Supplementary Table S5. Regarding the intensity patterns: we also wrongly typed the information for LFA-1 (ITGAL) in Figure S17. The correct pattern is d-d-d-u. The situation has now been corrected and we apologize for our carelessness. The 2000+ theoretical models (64 when simplified into x-x-x-x type patterns) generated do not cover every possible

	intensity values myself because they were not provided to me (see comment 3). In SF17, they reported on a few of the proteins in each of these groups, but the classes they reported did not seem to visually match up with the intensities at all. Eg: LFA-1, 0-1min:WT says down, but if anything it is slightly up? More likely constant. This is repeated throughout the figures. The authors also report on changes observed between incredibly low intensities (<1000), which I find very problematic to have confidence in.	scenario. If all possible models were generated, the total numbers would be 3^4 or 81. Many “wavy” models such as u-d-u-d were not considered to save effort, and because they were considered very rare. Each remaining theoretical pattern competes for the winning status and the pattern with the smallest distance to the intensity data from a given protein is assigned to that protein. Furthermore, generally speaking, the algorithm employed in calculating the patterns is purposely somewhat immune to noise, and slight up or down at any given interval can be interpreted as “constant”, especially if the following interval goes in the opposite direction. On the other hand, a slight up (or down) that looks like a near-constant, may be interpreted as up (or down) if the following intervals go also up (or down). For these reasons, the algorithmically calculated pattern may not in all cases be what we see by eye. We appreciate the concern regarding the low intensities of some of the detected proteins. Our replicate data is very consistent and reproducible, even for the low-intensity proteins, and we thus have confidence in our data. We have included the protein intensities in Supplementary Tables.
8	In Figure 5B, the Y axis has been cut off. Both TMEM16F and PIEZO1 have very low intensity values in WT, so I would not say its fair to report those proteins as ‘missing’ from the extracellular exposed proteome because they are hardly detected in the WT samples.	We thank the reviewer for pointing out our mistake during the figure composition. It has been now fixed. Our intent was never to state or imply that the proteins are missing (except for TMEM16F in the KO cells, validating the genetic data), just that they are not detected. One tidbit regarding scramblase and channel proteins such as TMEM16F and PIEZO1 is that their surface expression is intrinsically much lower than receptors or adhesion proteins such that antibody-based detection may not be sensitive even after signal amplification. In the case of both

		TMEM16F and PIEZO1 proteins, since their riBAQ signals are consistent and reproducible in all three replicates, we are confident in our data.
9	In the discussion of the results, the authors provided a lot of supposition about what their findings could suggest about TMEM16F functions. However, they did not test any of these hypotheses. Indeed as with most omics studies, there are a lot of possible biological changes that can be followed- up on, and not all can be carried out, but by not following up on any of the hypotheses generated by the author's interpretation of the results, I find it challenging to have confidence in the method itself being able to successfully report on new biology. The sheer number of specific interpretations of the results without evidence is very problematic to me, especially when there is concern about the accuracy of the technique itself.	We acknowledge the reviewer's concern about not having follow-up into any novel biological hypotheses. Short answer: we indeed uncovered a novel cellular pathway from the dataset that allows us to expand the TMEM16F biology and are actively pursuing that line of research at this moment. Given the novelty and amount of work required to support the hypothesis, however, we decided to exclude that finding in this manuscript since it does not further illuminate the validity of SLAPSHOT itself. We also wish to emphasize the technical aspects of SLAPSHOT rather than taking space to further scrutinize the TMEM16F biology so that readers are well-informed on the method itself and could garner as much information as possible in order to successfully employ SLAPSHOT for exploring the biological unknowns. All that said, we meticulously interpreted our data so that we could identify proteins that partake in a rather peculiar cellular phenomenon of TMEM16F-mediated surface area expansion and related MEND processes. It is noteworthy that, prior to SLAPSHOT study, it is not clear what proteins are found on the newly added membrane due to the limitation of the temporal sensitivity of available methods. With SLAPSHOT, we were able to assign proteins that matched cellular morphological changes recorded by others. Unfortunately, such an interpretation is not possible to easily validate using classical methods due to the limitations in their temporal resolution. Additionally, given the vast and unknown physiological implications of the cellular phenomenon

		itself, it would not be easy to design, perform and complete such experiments within a reasonable time frame. That said, we hope that our data could be a primer for further investigations into this phenomenon. As for the accuracy of the technique, we wish to reiterate that, SLAPSHOT is not a method for mapping plasma membrane proteins. Rather, SLAPSHOT is meant to capture extracellularly exposed proteome (including many canonical soluble proteins that are exposed to the extracellular space via different routes) and is tremendously useful if one wishes to study fleeting changes in cellular events.
--	--	--

Reviewer #2 (Remarks to the Author):

The authors present a novel enzymatic labeling method for proteomic analysis of the surfaceome, alternative to Cell Surface Capture or lysine biotinylation using sulfo-NHS-biotin. The presented benefit of this method lies in the possibility of tracking temporal dynamics of the proteome thanks to the labeling efficiency at low temperatures. The article contains details of the optimization procedure and a comparison with the established methods for surface labeling. The core of this work lies in the comparative quantitative time-course proteomic analysis of wild-type cells and cells lacking TMEM16F in response to calcium stimulation, performed using the SLAPSHOT method. The method is well suited for the biological issue under scrutiny and could be employed in a broad spectrum of biological issues.

The article is robust yet clear overall, and the amount of work done is impressive. Yet, I have several specific recommendations and also a few doubts, which I believe may be refuted by the authors:

	Comments	Response
1	Some changes that should be made in the Introduction and beginning of the Results for more clarity: Comparison of the SLAPSHOT method with the established surface labeling approaches is made in the results. The discussion then contains arguments further comparing these methods. The	We thank the reviewer for their comments. We have modified the introduction to introduce the enzyme, the reagents, SLAPSHOT's status as an alternative for the existing methods earlier for the reader's convenience. (lines 56, 59, and 92)

	statement that this method presents an alternative to these methods should also be mentioned in the introduction. The introduction omits information about which molecule specifically is used for enzymatic labeling. This information first appears in the results section, even the fact that the method uses biotinylation first appears in the results. Similarly, the fact that the method uses the label BxxT is mentioned (p. 5) before it is actually explained (p. 6) what this molecule is and why it is used.	
2	Figure 1B is problematic. The data presented in Figure 1B, the part stained with Streptavidin-HRP, are not informative, as the signal intensities vary among the different methods. The differences between soluble fraction and pellet are incomparable, as some “S” lanes are likely below the detection limit. The figure would be informative if parts of the blot for each of the methods were detected separately and the intensity of the “P” lanes visually at least approximately matched. Moreover, this figure is supposed to demonstrate that the biotinylation in the SLAPSHOT method is mostly restricted to the membrane fraction. However, I am not sure I can assess if a correct approach was made since the related section in the Methods (“Other labeling methods and hypotonic crude fractionation,” p. 22) cites “Teo and Wells, 2014”, which does not contain any information about the method used for hypotonic cell lysis, or fractionation leading to isolation of membrane fraction, nor does any other section of the Methods.	While we understand the reviewer’s point of view, we would like to explain our intention since we think there is more than one way of designing the streptavidin blot experiment depending on one’s focus. The primary question we asked in the experiment was “what was the overall biotinylation level of SLAPSHOT compared to two other methods”; and followed by “could SLAPSHOT preferentially label plasma membrane proteins as we predicted (given that both the enzyme the key substrate BxxT are membrane impermeable)”. Hence, we decided to place the emphasis on comparing overall biotinylation levels in the same amount of lysate starting from the same amount of cells in parallel experiments. The soluble (cytoplasmic) and pellet (plasma membrane enriched) fractions were obtained from cells lysed upon 60 min hypotonic swelling on ice. After centrifugation at 22,000g for 30 min, we assigned the supernatant as “soluble” and the pellet as “crude membrane”. We apologize for overlooking the details and using the wrong reference where the method was adapted from. We have made corrections to the method section and reference list accordingly (line 552).

3	The explanation of the presence of intracellular soluble proteins in the SLAPSHOT analysis. The comparison of SLAPSHOT and CSC (p. 6 and p. 7) shows that CSC is somewhat more specific in identifying plasma membrane proteins, while SLAPSHOT identifies abundance of intracellular soluble proteins. This may point either to a flaw in the method's design or to the explanation about the secretome presented by the authors. The microscopic data (p.7-8, figure S6) point to the integrity of the membrane, therefore, label permeability and membrane rupture should not be the issue. However, the presence of His-APEX2 in the washed samples (p. 15, figure S16) leads me to a suspicion that labeling may continue during the capture procedure. Can you rule this suspicion out?	We understand reviewer's concern since we had the same concerns when we were planning our experiments, Thus, we took many steps to inhibit APEX2 activity after the intended labeling period: (1) we quenched the reaction with known and potent APEX2 inhibitors (data from us and the Ting lab), (2) Cells were washed to remove water-soluble materials, and the same quenchers were added to the wash buffer as well, (3) The same quenchers were present in the lysis buffer as well. Furthermore, sodium azide is an irreversible heme-specific inhibitor, so it will effectively inhibit APEX2 upon contact. APEX2 is a sticky protein and some of it is carried throughout the sample manipulations all the way to the mass spectrometer, but based on the quenching assays we are confident that the enzyme is inactivated already during the first quenching immediately post-labeling.
4	Data support for the secretome explanation. "Notably, these 1679 proteins with intracellular annotation include 1509 (89.9%) that have been detected in the secretome and/or the extracellular vesicle proteome of U2OS cells" (p. 7) – as this is a crucial argument for explaining why are these proteins identified in such abundance when using SLAPSHOT, the overlap should be shown, e.g. in an additional supplementary table. Does CSC detect any of such proteins?	We believe this discrepancy is amplified by the intrinsic properties of SLAPSHOT and CSC targets. CSC labels cis-diol functionality that is found predominantly on the glycans of plasma membrane and classical extracellular secreted proteins since they ought to go through the glycosylation machinery (on the secretory pathway) before reaching the plasma membrane/extracellular environment. Thus, CSC labeling precludes any intracellular proteins that lack glycans (i.e. most of the intracellular proteins). For a typical glycoprotein, it usually possesses multiple biotinylation sites per one polypeptide (one cis-diol from a single carbohydrate can accept two biotins; majority of the glycans bear more than one cis-diol-containing carbohydrates). SLAPSHOT, on the other hand, does not have such prerequisite since its targets are

		the side chains on tyrosine and tryptophan residues. The intrinsic low availability of Tyr and Trp on the polypeptide backbone (as shown in Supplementary Fig. 6) does not guarantee that every plasma membrane protein expressed on the cells can easily be labeled by SLAPSHOT (also, plasma membrane proteins may have relatively small extracellular loops further diminishing the chance of Tyr and Trp being presented extracellularly). Collectively, these scenarios account for our proteomic data in which CSC outperformed SLAPSHOT in identified known plasma membrane and canonical secreted proteins, most of which are glycosylated. As for the presence of all the canonical intracellular proteins found in SLAPSHOT samples, after all the control experiments, we think these proteins are truly extracellularly exposed, albeit in this study we were not able to pinpoint the mode of their arrival. While we tend to imagine the plasma membrane as a rigid barrier, we often ignore the myriad activities that a living cell constantly undergoes. One aspect is the constant release of intracellular materials via various routes. Especially, for many of the intracellular proteins, their extracellular exposed status is not static (i.e. their intensity change in response to ionomycin stimulation etc.) suggesting that they are indeed present. We have added a new Supplementary Table 2 to indicate the number of these proteins detected by the two methods. CSC also detected 1120 intracellular annotated proteins, and 1094 of which are also found in SLAPSHOT (line 143).
5	Limitations of the method may be discussed in the Discussion section. For example, “low availability of the phenoxy radical targets in the	We have changed the text to discuss these limitations of SLAPSHOT (line 379).

	proteome” (p. 7) seems like a potential limitation. However, even though it is mentioned in the discussion (p. 13), it is not presented as a limitation. Another example may be “our initial MS analysis detected ... unique set of proteins with plasma membrane annotation that were not detected by CSC” (p. 17) – there were 3 times more unique plasma membrane proteins detected by CSC. This seems like a limitation worth mentioning.	
6	A potential inconsistency of the text in Results with the data presented in Figure 4B. “(f) Curiously the c-c-u-u pattern in 16F KO cells only contains five proteins but collectively accounts for higher riBAQ value than many other patterns with more protein constituents (Figure 6E).” (p.13) – Figure 4B displays the riBAQ values using the color gradient, but it is not visible that the c-c-u-u pattern in 16F KO cells has any high values. Therefore, it is unclear to me whether this claim is wrongly formulated or mistaken.	We thank the reviewer for going through the figures so carefully and apologize for the error. The text was correctly written, and the figure had an error in it. The mistake has now been corrected.
7	A dubious statement in the Discussion. “... Such phenomenon further confirms that mitovesicle release is insensitive to acute intracellular calcium elevation ...” (p. 13) – the increase in mitovesicles happens in both cell lines subjected to Ca influx. Yes, they do not react immediately, but on what do you base the claim that they are insensitive to Ca influx? Additionally, I would not call the different temporal pattern “drastically different” (p. 17)	We agree with the reviewer that this sentence was not clear. Our intent was to convey the message that mitovesicle release is TMEM16F-and/or calcium-independent because their patterns do not match in WT and 16FKO cells. Regarding the sentence on p.17, we agree with the sentiment, this was misinterpreted by us. Both sentences were removed in efforts to reduce the overall length of the manuscript.
8	Further minor details.	We are thankful for the reviewer pointing out these mistakes.
8-1	Introduction: (p. 3) The abbreviation PECSL is not explained.	Fixed (line 62)

8-2	Results: (p. 5) “(see Materials and Methods)” – the reference should be more specific	Fixed (line 130)
8-3	(p. 7) Supplementary figure A and B – in all cases in the text (figure description and p. 14), A refers to B in the figure and vice versa	Fixed
8-4	(p. 10) ““488 and 460 membrane proteins” – how are membrane proteins defined here, are they based on an annotation or prediction? Similarly, when mentioning annotations (p. 7), it should be stated, which precise annotations from which database were used	Fixed (lines 130 and 213)
8-5	(p. 12) “(Figure 5C)” – should be 6C.	Fixed (line 282)
8-6	(p. 12) “...is the second most populated...” – is actually third, when counting the c-c-c-c pattern	Fixed (line 283)
8-7	Figures: (p. 35) The bitmap in 5A covers the top of the graphs in 5B. Also, there is an object in the bottom right corner of the bitmap in 5A that should be cropped out	Fixed

Reviewers' comments:

Reviewer #1 (Remarks to the Author):

The authors have updated the manuscript to make it more concise and direct. They have corrected the erroneous figures that both reviewers were able to catch. They have also now included links/charts for their MS data. However, they have not performed any further experiments from the first submission to convincingly show that the method is successfully labeling only extracellularly-exposed proteins. Their claims require further experimental evidence to back up their hypothesis (outline in their rebuttal) as to how these intracellular proteins are being labeled in line with their method working properly, to refute the obvious conclusion that the method is not working as stated. Confidence in this method capturing true, unusual extracellularly-exposed proteins requires follow up of some of the low intensity hits with additional experiments: IF showing recruitment of protein X to membrane, membrane fractionation experiments, capturing exosomes and using MS to ID contents, reciprocal KO experiments, etc.. Without further evidence, I am unconvinced this method works as stated, or that the low intensity hits identified are bona fide, issues that remain critical flaws of the manuscript.

Reviewer #2 (Remarks to the Author):

The authors have adequately addressed all of my previous comments. They have made the necessary corrections and have provided me with satisfactory explanations for the concepts I had concerns about. The clarifications made in the introduction, along with the reduction in text length, have greatly improved the manuscript's comprehensibility.

To the editors and reviewers:

We thank everyone for their input. Below **in blue** are the reviewer's comments, and **in red** are our amended comments **accompanying the second revision** of the manuscript.

Reviewer #1 (Remarks to the Author):

The authors have updated the manuscript to make it more concise and direct. They have corrected the erroneous figures that both reviewers were able to catch. They have also now included links/charts for their MS data. However, they have not performed any further experiments from the first submission to convincingly show that the method is successfully labeling only extracellularly-exposed proteins. Their claims require further experimental evidence to back up their hypothesis (outline in their rebuttal) as to how these intracellular proteins are being labeled in line with their method working properly, to refute the obvious conclusion that the method is not working as stated. Confidence in this method capturing true, unusual extracellularly-exposed proteins requires follow up of some of the low intensity hits with additional experiments: IF showing recruitment of protein X to membrane, membrane fractionation experiments, capturing exosomes and using MS to ID contents, reciprocal KO experiments, etc.. Without further evidence, I am unconvinced this method works as stated, or that the low intensity hits identified are bona fide, issues that remain critical flaws of the manuscript.

Reviewer #2 (Remarks to the Author):

The authors have adequately addressed all of my previous comments. They have made the necessary corrections and have provided me with satisfactory explanations for the concepts I had concerns about. The clarifications made in the introduction, along with the reduction in text length, have greatly improved the manuscript's comprehensibility.

We thank both reviewers for their efforts.

We have performed isolation and analysis of extracellular vesicles (EVs), as suggested by Reviewer #1. We used western blotting to analyze EVs isolated from Jurkat cells at multiple time points as well as pellets of the remaining Jurkat cells after the time-course ionomycin stimulation regiment for assessment of released EVs. Our new data provide further indication that SLAPSHOT does not indiscriminately label cellular contents for the following reasons: (1) Diacylglycerol O-acyltransferase 2 (DGAT2), an ER protein that has never been reported to be in EVs (as per Vesiclepedia), was not labeled by SLAPSHOT. (2) A substantial portion of cellular TMEM16F is found in the EVs themselves, a phenomenon that is consistent with its intimate role in the EV shedding. (3) This western blot analysis provides validation for several quantitative differences that our SLAPSHOT analysis revealed between Jurkat cells with or without TMEM16F. All the western blotting data (including densitometric quantitation) are presented in the Supplementary Figure 18. The major textual changes regarding the new data can be found in lines 302 and 364 of the revised manuscript. Materials and Methods section was modified accordingly (starting at line 657 and 673).

On a final note, we also appended an extra tab in the Supplementary Data Excel file where the readers can reproduce all time-course plots presented in the manuscript, and gave a proper name for a tab (Supplementary Table 7, with a citation in the paper at line 795) was left earlier unnamed.

General remarks to the editor and both reviewers:

We sincerely thank both reviewers for their thoughtful and valuable criticisms. We have made adjustments and corrections to the manuscript accordingly and also addressed the issues raised in an itemized table as shown below. As a general note, we have condensed the manuscript to adhere to the journal guidelines. Furthermore, we have added several more supplementary tables, also to adhere to the journal guidelines, to show the underlying data for each figure. Finally, we have modified Supplementary Figure 13 to properly illustrate the data in more detail.

Reviewer #1 (Remarks to the Author):

In their manuscript, Tuomivaara et al report on SLAPSHOT, a new proximity proteomics method to identify the extracellularly exposed proteome in live cells. This technique relies on modification of the APEX method to allow for investigation of endogenous proteins, with no requirement for cell or protein engineering. After demonstrating optimization efforts for the use of SLAPSHOT in cells, the authors apply this strategy to understand dynamic changes in the extracellularly exposed proteome of Jurkat cells in response to CRISPR-mediated deletion of the calcium-dependent phospholipid scramblase and ion channel protein TMEM16F. These studies demonstrated differences in proteins identified based on TMEM16F expression, with WT cells displaying more proteins from the ER and KO cells displaying more proteins from the mitochondria. Although the authors have identified an important need for technologies that facilitate labelling of the cell surface proteome to answer important questions like the function of TMEM16F in regulating it, I have identified several concerns with this study in its current form:

	Comments	Responses
1	The manuscript is very long and would benefit from more focused, concise reporting of background information, the results and discussion. In particular, the discussion was abnormally lengthy and seemed to re-iterate the findings more than position the results within the field. That being said, the authors did well in referencing the current literature.	We appreciate the suggestions and the compliment. We have condensed all parts of the manuscript significantly for the readers' convenience and to adhere to the journal word limit. We truncated the word count from 6088 to 4501.
2	There are blanks/highlighted areas where data deposition information should have been input. This is essential for an open peer review process.	We apologize for leaving out the data deposition details. The situation has now been corrected (lines 772 and 776). The reviewers can access the proteomics data at the PRIDE repository with dataset identifier PXD041387 with username:

		reviewer_pxd041387@ebi.ac.uk and password: 7gc1QBUD. The plasmid ID at Addgene is 201471. The plasmid and MS data will be released for public access upon publication of this manuscript.
3	I also was surprised that no table was included showing the individual intensity data for each gene identified across the WT and KO replicates. This is usually provided with the paper as an excel file. The file that was included seems to compare the 5 replicates that were used for the initial optimization of the SLAPSHOT method. However, even here the authors only report on ‘detected’ or not. This doesn’t give the reader any information on the intensity of the detection or what the consistency across runs is like. There is no table included that compares identifications for WT vs TMEM16F KO.	We agree that this information is valuable and should be reported. Tables with intensity information are included for both the method development part with U2OS cells (appended in Supplementary Table 1, as well as the calcium stimulation experiment with Jurkat cells (new Supplementary Table 3). The tables include intensity information for all replicates.
4	From what I gather, the authors used MS1 quantification of DDA data, although this was not implicitly stated. This is still the most reported quantification method in the field, however there is a movement towards DIA quantification methods, that are less stochastic than those that involve selection of a limited number of precursor peptides for fragmentation. At minimum I would comment on the limitations of this ‘quantification’ method.	A comment regarding the limitations of DDA experiments has been added to the Discussion (line 399).
5	The authors did a good job of positioning the unique utility of their technique in comparison to the wave of similar methods that have recently been reported on. While PECSL (Li et al., 2021) is a similar strategy using HRP, other APEX-based strategies involve protein engineering to anchor the enzyme to the cell surface (Li et al., 2020; Kirkemo et al., 2022). The uniqueness of their strategy would seem to shine most in its	We thank the reviewer’s kind words. We agree that SLAPSHOT would be a useful approach for tissue or other samples that are recalcitrant for genetic manipulation (as mentioned in the discussion). At the inception of this work, we identified two aspects in the extracellularly exposed proteome mapping efforts that SLAPSHOT could have advantage at: (1) the temporal advantage where fleeting changes during fast signaling events can be captured; (2)

	use for systems where protein engineering is not possible, such as in tissues; however, they chose to use it in cell systems, where I think there is less novelty in the method, and they are unable to demonstrate a clear advancement using their method. I would also like to note that the authors indicated APEX is ideal versus other strategies because it targets Tyrosine and Tryptophans and not Lysines, and that targeting lysines can interfere with proteolysis at later steps of processing. Although theoretically logical and there are some studies reporting this to be the case, there is rather weak evidence of this actually negatively impacting identifications in practice, and I do not think it should be highlighted as a major advantage of the SLAPSHOT method.
6	The authors compared their method to the classical CSC method and while they captured significantly more proteins in significantly less time, the novel proteins identified were mostly enriched for those that are not typically described on the cell surface. This suggests that the methodology may require further development rather than being superior as the authors present it. Although there was not evidence of labelling in the interior of fixed cells, there remains the possibility that labelling is occurring post lysis using this method, leading to a large increase of intracellular protein labeling. Additional controls may assist in teasing this out, but as it stands, I am not confident that this method truly labels only the extracellular exposed proteome in live cells. The authors surmise that some of these identifications could be related to exosomes from the ER or mitochondria, but based on the methods they've described, where successive washing is performed prior to cell lysis, I samples whose genetic manipulation is nontrivial. After careful consideration, we decided to focus on showcasing the temporal advantage, since there are many clever approaches and technological advancements in addressing the second aspect. That said, we look forward to seeing SLAPSHOT in tissue labeling. We have toned down the section where we discussed the biotinylated lysine effect on trypsin digestion (line 306). We recognize the reviewer's concerns regarding the labeling of canonical intracellular proteins with SLAPSHOT, for we had the same thoughts upon inspecting the proteomics data. As pointed out by the reviewer, microscopic and western blot evidence supports the spatial specificity of SLAPSHOT. After experimenting and testing various possibilities in order to get to the bottom of the matter (as shown in the supplementary materials), we utilized Occam's razor and concluded that these canonical intracellular proteins are indeed exposed to the extracellular milieu. The strength of the classical CSC in surfaceome mapping is that its chemical reaction targets the complex glycans on the plasma membrane and many canonical extracellularly secreted proteins. That said, in our dataset, CSC method also identified more than 1000 canonical intracellular proteins, likely due to the presence of O-GlcNAcylation (an abundant intracellular glycosylation) and carbonylation (another naturally occurred but underappreciated

wouldn't expect exosomes to have been collected and thereby identified by MS.	PTM that is a directed target for hydrazide-biotin) proteins. Unlike complex glycosylation, O-GlcNAcylation and carbonylation display low stoichiometry. Thus, these intracellular proteins are often considered noise and have never been seriously considered in most surfaceome mapping efforts. Also, depending on the scientific question (e.g. if one is only interested on plasma membrane surface biomarker), they are often simply excluded from the discussion. SLAPSHOT targets polypeptide directly (tyrosine and tryptophan side chains) rather than PTMs. Thus, it does not discriminate the origin of the proteins, as long as they are extracellularly accessible. We agree with the reviewer that vesicles are likely washed away during the sample processing. Our hypothesis is that many of the “released vesicle proteins” are en route to the extracellular space before they have a chance to fully separate from the cells and are labeled and captured due to the fast kinetic afforded by SLAPSHOT. The observation of such phenomenon in vesicle shedding can be found in the videos in several publications, such as PMID: 30622179; 31995754. We also would like to emphasize that we do not claim that SLAPSHOT labeling is specific for surfaceome (which often equates to plasma membrane residents). Rather, we present SLAPSHOT as a method to capture extracellularly exposed proteins. Additionally, we are not claiming that SLAPSHOT is superior to CSC (although its time resolution is). Rather, we stated that SLAPSHOT could be applied to address certain biological questions that might not be answerable using other methods. We believe all methods have their unique strengths along with shortcomings.
--	--

		Thus, one should consider all aspects of the scientific question at hand and carefully choose an appropriate tool from the “toolbox”. We appreciate the comment regarding labeling after the intended labeling period, and this is one of the issues we wanted to rigorously rule out starting from the initial sketching of the experiments. Our experimental design makes post-lysis labeling very unlikely since: (1) The labeling reaction was quenched with known and potent inhibitors (as shown in the supplementary materials as well as the Ting group in their publications) immediately after the labeling period, (2) The cells were washed prior to lysis, removing all water-soluble reagents, also with the quenchers included in the wash buffers, neutralizing any residual reactive enzyme molecules. (3) Lysis buffer itself contains the same quenchers. We also note that sodium azide is an irreversible heme-specific inhibitor, so it will effectively inhibit APEX2 upon contact.
7	The authors present the timecourse data not as intensity values for each protein identified, but rather based on its identification as being increased (u), decreased (d) or constant (c) as compared to the intensity value in the previous timepoint. Although these comparisons are important, I found this binning method needlessly confusing. And when you actually look at the intensity values for the select proteins that they did present in their more raw form (SF17), they did not see to match the binning value at all. For example, from the heatmap in figure 4, the intensities observed across the d-u-u-u subgroup seem to have very similar intensities across the groups, and I would have	We thank the reviewer for inspecting our data with a fine comb. Figure 4 was wrongly assembled and has now been corrected. The intensities in the heatmap are still somewhat subtle, as most proteins have the c-c-c-c pattern, blunting the visual changes relative to the total intensity. We also included the intensity pattern for each protein in a new Supplementary Table S5. Regarding the intensity patterns: we also wrongly typed the information for LFA-1 (ITGAL) in Figure S17. The correct pattern is d-d-d-u. The situation has now been corrected and we apologize for our carelessness.

expected them to be labelled c-c-c-c instead? I could not actually check the intensity values myself because they were not provided to me (see comment 3). In SF17, they reported on a few of the proteins in each of these groups, but the classes they reported did not seem to visually match up with the intensities at all. Eg: LFA-1, 0-1min:WT says down, but if anything it is slightly up? More likely constant. This is repeated throughout the figures. The authors also report on changes observed between incredibly low intensities (<1000), which I find very problematic to have confidence in.	The 2000+ theoretical models (64 when simplified into x-x-x-x type patterns) generated do not cover every possible scenario. If all possible models were generated, the total numbers would be 3^4 or 81. Many “wavy” models such as u-d-u-d were not considered to save effort, and because they were considered very rare. Each remaining theoretical pattern competes for the winning status and the pattern with the smallest distance to the intensity data from a given protein is assigned to that protein. Furthermore, generally speaking, the algorithm employed in calculating the patterns is purposely somewhat immune to noise, and slight up or down at any given interval can be interpreted as “constant”, especially if the following interval goes in the opposite direction. On the other hand, a slight up (or down) that looks like a near-constant, may be interpreted as up (or down) if the following intervals go also up (or down). For these reasons, the algorithmically calculated pattern may not in all cases be what we see by eye. We appreciate the concern regarding the low intensities of some of the detected proteins. Our replicate data is very consistent and reproducible, even for the low-intensity proteins, and we thus have confidence in our data. We have included the protein intensities in Supplementary Tables.
8 In Figure 5B, the Y axis has been cut off. Both TMEM16F and PIEZO1 have very low intensity values in WT, so I would not say its fair to report those proteins as ‘missing’ from the extracellular exposed proteome because they are hardly detected in the WT samples.	We thank the reviewer for pointing out our mistake during the figure composition. It has been now fixed. Our intent was never to state or imply that the proteins are missing (except for TMEM16F in the KO cells, validating the genetic data), just that they are not detected. One tidbit regarding scramblase and channel proteins such as TMEM16F and PIEZO1 is that their surface expression is intrinsically much lower than receptors or

		adhesion proteins such that antibody-based detection may not be sensitive even after signal amplification. In the case of both TMEM16F and PIEZO1 proteins, since their riBAQ signals are consistent and reproducible in all three replicates, we are confident in our data.
9	In the discussion of the results, the authors provided a lot of supposition about what their findings could suggest about TMEM16F functions. However, they did not test any of these hypotheses. Indeed as with most omics studies, there are a lot of possible biological changes that can be followed- up on, and not all can be carried out, but by not following up on any of the hypotheses generated by the author's interpretation of the results, I find it challenging to have confidence in the method itself being able to successfully report on new biology. The sheer number of specific interpretations of the results without evidence is very problematic to me, especially when there is concern about the accuracy of the technique itself.	We acknowledge the reviewer's concern about not having follow-up into any novel biological hypotheses. Short answer: we indeed uncovered a novel cellular pathway from the dataset that allows us to expand the TMEM16F biology and are actively pursuing that line of research at this moment. Given the novelty and amount of work required to support the hypothesis, however, we decided to exclude that finding in this manuscript since it does not further illuminate the validity of SLAPSHOT itself. We also wish to emphasize the technical aspects of SLAPSHOT rather than taking space to further scrutinize the TMEM16F biology so that readers are well-informed on the method itself and could garner as much information as possible in order to successfully employ SLAPSHOT for exploring the biological unknowns. All that said, we meticulously interpreted our data so that we could identify proteins that partake in a rather peculiar cellular phenomenon of TMEM16F-mediated surface area expansion and related MEND processes. It is noteworthy that, prior to SLAPSHOT study, it is not clear what proteins are found on the newly added membrane due to the limitation of the temporal sensitivity of available methods. With SLAPSHOT, we were able to assign proteins that matched cellular morphological changes recorded by others. Unfortunately, such an interpretation is not possible to easily validate using classical methods due to the limitations in their

		temporal resolution. Additionally, given the vast and unknown physiological implications of the cellular phenomenon itself, it would not be easy to design, perform and complete such experiments within a reasonable time frame. That said, we hope that our data could be a primer for further investigations into this phenomenon. As for the accuracy of the technique, we wish to reiterate that, SLAPSHOT is not a method for mapping plasma membrane proteins. Rather, SLAPSHOT is meant to capture extracellularly exposed proteome (including many canonical soluble proteins that are exposed to the extracellular space via different routes) and is tremendously useful if one wishes to study fleeting changes in cellular events.
--	--	--

Reviewer #2 (Remarks to the Author):

The authors present a novel enzymatic labeling method for proteomic analysis of the surfaceome, alternative to Cell Surface Capture or lysine biotinylation using sulfo-NHS-biotin. The presented benefit of this method lies in the possibility of tracking temporal dynamics of the proteome thanks to the labeling efficiency at low temperatures. The article contains details of the optimization procedure and a comparison with the established methods for surface labeling. The core of this work lies in the comparative quantitative time-course proteomic analysis of wild-type cells and cells lacking TMEM16F in response to calcium stimulation, performed using the SLAPSHOT method. The method is well suited for the biological issue under scrutiny and could be employed in a broad spectrum of biological issues.

The article is robust yet clear overall, and the amount of work done is impressive. Yet, I have several specific recommendations and also a few doubts, which I believe may be refuted by the authors:

	Comments	Response
1	Some changes that should be made in the Introduction and beginning of the Results for more clarity: Comparison of the SLAPSHOT method with the established surface labeling	We thank the reviewer for their comments. We have modified the introduction to introduce the enzyme, the reagents, SLAPSHOT's status as an alternative for the

	approaches is made in the results. The discussion then contains arguments further comparing these methods. The statement that this method presents an alternative to these methods should also be mentioned in the introduction. The introduction omits information about which molecule specifically is used for enzymatic labeling. This information first appears in the results section, even the fact that the method uses biotinylation first appears in the results. Similarly, the fact that the method uses the label BxxT is mentioned (p. 5) before it is actually explained (p. 6) what this molecule is and why it is used.	existing methods earlier for the reader's convenience. (lines 56, 59, and 92)
2	Figure 1B is problematic. The data presented in Figure 1B, the part stained with Streptavidin-HRP, are not informative, as the signal intensities vary among the different methods. The differences between soluble fraction and pellet are incomparable, as some "S" lanes are likely below the detection limit. The figure would be informative if parts of the blot for each of the methods were detected separately and the intensity of the "P" lanes visually at least approximately matched. Moreover, this figure is supposed to demonstrate that the biotinylation in the SLAPSHOT method is mostly restricted to the membrane fraction. However, I am not sure I can assess if a correct approach was made since the related section in the Methods ("Other labeling methods and hypotonic crude fractionation," p. 22) cites "Teo and Wells, 2014", which does not contain any information about the method used for hypotonic cell lysis, or fractionation leading to isolation of membrane fraction, nor does any other section of the Methods.	While we understand the reviewer's point of view, we would like to explain our intention since we think there is more than one way of designing the streptavidin blot experiment depending on one's focus. The primary question we asked in the experiment was "what was the overall biotinylation level of SLAPSHOT compared to two other methods"; and followed by "could SLAPSHOT preferentially label plasma membrane proteins as we predicted (given that both the enzyme the key substrate BxxT are membrane impermeable)". Hence, we decided to place the emphasis on comparing overall biotinylation levels in the same amount of lysate starting from the same amount of cells in parallel experiments. The soluble (cytoplasmic) and pellet (plasma membrane enriched) fractions were obtained from cells lysed upon 60 min hypotonic swelling on ice. After centrifugation at 22,000g for 30 min, we assigned the supernatant as "soluble" and the pellet as "crude membrane". We apologize for overlooking the details and using the wrong reference where the method

		was adapted from. We have made corrections to the method section and reference list accordingly (line 552).
3	The explanation of the presence of intracellular soluble proteins in the SLAPSHOT analysis. The comparison of SLAPSHOT and CSC (p. 6 and p. 7) shows that CSC is somewhat more specific in identifying plasma membrane proteins, while SLAPSHOT identifies abundance of intracellular soluble proteins. This may point either to a flaw in the method's design or to the explanation about the secretome presented by the authors. The microscopic data (p.7-8, figure S6) point to the integrity of the membrane, therefore, label permeability and membrane rupture should not be the issue. However, the presence of His-APEX2 in the washed samples (p. 15, figure S16) leads me to a suspicion that labeling may continue during the capture procedure. Can you rule this suspicion out?	We understand reviewer's concern since we had the same concerns when we were planning our experiments, Thus, we took many steps to inhibit APEX2 activity after the intended labeling period: (1) we quenched the reaction with known and potent APEX2 inhibitors (data from us and the Ting lab), (2) Cells were washed to remove water-soluble materials, and the same quenchers were added to the wash buffer as well, (3) The same quenchers were present in the lysis buffer as well. Furthermore, sodium azide is an irreversible heme-specific inhibitor, so it will effectively inhibit APEX2 upon contact. APEX2 is a sticky protein and some of it is carried throughout the sample manipulations all the way to the mass spectrometer, but based on the quenching assays we are confident that the enzyme is inactivated already during the first quenching immediately post-labeling.
4	Data support for the secretome explanation. "Notably, these 1679 proteins with intracellular annotation include 1509 (89.9%) that have been detected in the secretome and/or the extracellular vesicle proteome of U2OS cells" (p. 7) – as this is a crucial argument for explaining why are these proteins identified in such abundance when using SLAPSHOT, the overlap should be shown, e.g. in an additional supplementary table. Does CSC detect any of such proteins?	We believe this discrepancy is amplified by the intrinsic properties of SLAPSHOT and CSC targets. CSC labels cis-diol functionality that is found predominantly on the glycans of plasma membrane and classical extracellular secreted proteins since they ought to go through the glycosylation machinery (on the secretory pathway) before reaching the plasma membrane/extracellular environment. Thus, CSC labeling precludes any intracellular proteins that lack glycans (i.e. most of the intracellular proteins). For a typical glycoprotein, it usually possesses multiple biotinylation sites per one polypeptide (one cis-diol from a single carbohydrate can accept two biotins; majority of the glycans bear more than one cis-diol-containing carbohydrates).

SLAPSHOT, on the other hand, does not have such prerequisite since its targets are the side chains on tyrosine and tryptophan residues. The intrinsic low availability of Tyr and Trp on the polypeptide backbone (as shown in Supplementary Fig. 6) does not guarantee that every plasma membrane protein expressed on the cells can easily be labeled by SLAPSHOT (also, plasma membrane proteins may have relatively small extracellular loops further diminishing the chance of Tyr and Trp being presented extracellularly).

Collectively, these scenarios account for our proteomic data in which CSC outperformed SLAPSHOT in identified known plasma membrane and canonical secreted proteins, most of which are glycosylated.

As for the presence of all the canonical intracellular proteins found in SLAPSHOT samples, after all the control experiments, we think these proteins are truly extracellularly exposed, albeit in this study we were not able to pinpoint the mode of their arrival. While we tend to imagine the plasma membrane as a rigid barrier, we often ignore the myriad activities that a living cell constantly undergoes. One aspect is the constant release of intracellular materials via various routes. Especially, for many of the intracellular proteins, their extracellular exposed status is not static (i.e. their intensity change in response to ionomycin stimulation etc.) suggesting that they are indeed present.

We have added a new Supplementary Table 2 to indicate the number of these proteins detected by the two methods. CSC also detected 1120 intracellular annotated proteins, and 1094 of which are also found in SLAPSHOT (line 143).

5	Limitations of the method may be discussed in the Discussion section. For example, “low availability of the phenoxy radical targets in the proteome” (p. 7) seems like a potential limitation. However, even though it is mentioned in the discussion (p. 13), it is not presented as a limitation. Another example may be “our initial MS analysis detected ... unique set of proteins with plasma membrane annotation that were not detected by CSC” (p. 17) – there were 3 times more unique plasma membrane proteins detected by CSC. This seems like a limitation worth mentioning.	We have changed the text to discuss these limitations of SLAPSHOT (line 379).
6	A potential inconsistency of the text in Results with the data presented in Figure 4B. “(f) Curiously the c-c-u-u pattern in 16F KO cells only contains five proteins but collectively accounts for higher riBAQ value than many other patterns with more protein constituents (Figure 6E).” (p.13) – Figure 4B displays the riBAQ values using the color gradient, but it is not visible that the c-c-u-u pattern in 16F KO cells has any high values. Therefore, it is unclear to me whether this claim is wrongly formulated or mistaken.	We thank the reviewer for going through the figures so carefully and apologize for the error. The text was correctly written, and the figure had an error in it. The mistake has now been corrected.
7	A dubious statement in the Discussion. “... Such phenomenon further confirms that mitovesicle release is insensitive to acute intracellular calcium elevation ...” (p. 13) – the increase in mitovesicles happens in both cell lines subjected to Ca influx. Yes, they do not react immediately, but on what do you base the claim that they are insensitive to Ca influx? Additionally, I would not call the different temporal pattern “drastically different” (p. 17)	We agree with the reviewer that this sentence was not clear. Our intent was to convey the message that mitovesicle release is TMEM16F-and/or calcium-independent because their patterns do not match in WT and 16FKO cells. Regarding the sentence on p.17, we agree with the sentiment, this was misinterpreted by us. Both sentences were removed in efforts to reduce the overall length of the manuscript.

8	Further minor details.	We are thankful for the reviewer pointing out these mistakes.
8-1	Introduction: (p. 3) The abbreviation PECSL is not explained.	Fixed (line 62)
8-2	Results: (p. 5) “(see Materials and Methods)” – the reference should be more specific	Fixed (line 130)
8-3	(p. 7) Supplementary figure A and B – in all cases in the text (figure description and p. 14), A refers to B in the figure and vice versa	Fixed
8-4	(p. 10) ““488 and 460 membrane proteins” – how are membrane proteins defined here, are they based on an annotation or prediction? Similarly, when mentioning annotations (p. 7), it should be stated, which precise annotations from which database were used	Fixed (lines 130 and 213)
8-5	(p. 12) “(Figure 5C)” – should be 6C.	Fixed (line 282)
8-6	(p. 12) “...is the second most populated...” – is actually third, when counting the c-c-c-c pattern	Fixed (line 283)
8-7	Figures: (p. 35) The bitmap in 5A covers the top of the graphs in 5B. Also, there is an object in the bottom right corner of the bitmap in 5A that should be cropped out	Fixed